# Daily ensemble river discharge reforecasts and real-time forecasts from the operational Global Flood Awareness System

Shaun Harrigan[1], Ervin Zoster[1,2], Hannah Cloke[2,3,4,5], Peter Salamon[6] and Christel Prudhomme[1,7,8]

[1]Forecast Department, European Centre for Medium-Range Weather Forecasts (ECMWF), Reading, UK
[2]Department of Geography and Environmental Science, University of Reading, Reading, UK
[3]Department of Meteorology, University of Reading, Reading, UK
[4]Department of Earth Sciences, Uppsala University, Uppsala, Sweden
[5]Centre of Natural Hazards and Disaster Science, CNDS, Uppsala, Sweden
[6] European Commission, Joint Research Centre (JRC), Ispra, Italy
[7]Centre for Ecology and Hydrology (CEH), Wallingford, UK
[8]Department of Geography and Environment, University of Loughborough, Loughborough, UK

*Correspondence to*: Shaun Harrigan (shaun.harrigan@ecmwf.int)

**Abstract:** Operational global-scale hydrological forecasting systems are widely used to help manage hydrological extremes such as floods and droughts. The vast amounts of raw data that underpin forecast systems and the ability to generate information on forecast skill have, until now, not been publicly available. As part of the Global Flood Awareness System (GloFAS; https://www.globalfloods.eu/) service evolution, in this paper daily ensemble river discharge reforecasts and real-time forecast datasets are made free and openly available through the Copernicus Climate Change Service (C3S) Climate Data Store (CDS). They include real-time forecast data starting on 1 January 2020 updated operationally every day and a 20-year set of reforecasts and associated metadata, available through the dedicated GloFAS FTP service. This paper describes the model components and configuration used to generate the real-time river discharge forecasts and the reforecasts. An evaluation of ensemble forecast skill using the Continuous Ranked Probability Skill Score (CRPSS) was also undertaken for river points around the globe. Results show that GloFAS is skilful in over 93 % of catchments in the short- (1- to 3-days) and medium-range (5- to 15-days) against a persistence benchmark forecast, and skilful in over 80 % of catchments out to the extended-range (16- to 30-days) against a climatological benchmark forecast. However, the strength of skill varies considerably by location with GloFAS found to have no or negative skill at longer lead times in broad hydroclimatic regions in tropical Africa, western coast of South America, and catchments dominated by snow and ice in high northern latitudes. Forecast skill is summarised as a new headline skill score available as a new layer on the GloFAS forecast Web Map Viewer to aid user's interpretation and understanding of forecast quality.

## 1 Introduction

Hydrological extremes, such as floods and droughts, have severe negative socio-economic impacts and climate change is expected to alter their timing and magnitude (Blöschl et al., 2017, 2019; Ward et al., 2020). Since 1990, reported disasters have led to over 94 million people affected by flooding each year, and economic losses are estimated at around USD 260-

310 billion per year (UNDRR, 2015a). The need to reduce this risk has been identified under the Sendai Framework for

Disaster Risk Reduction (UNDRR, 2015b). One of the primary methods of achieving DRR and building resilience in society is through early warning of extreme events. There are now several centers producing global and continental scale hydrological forecasts operationally which are working to support national forecasting and decision making in the water sector (Emerton et al., 2016). In Europe, there is the European Flood Awareness System (EFAS; www.efas.eu; Thielen et al., (2009) and European Hydrological Predictions for the Environment (E-HYPE; https://hypeweb.smhi.se/; Donnelly et al.,

2016); in the US the Hydrologic Ensemble Forecast Service (HEPS; https://water.weather.gov/ahps/; Demargne et al., 2014) and in Australian the Flood Forecasting and Warning Service (FFWS; http://www.bom.gov.au/water/). There are three global-scale systems: the Global Flood Awareness System (GloFAS; https://www.globalfloods.eu/, last accessed 23 September; Alfieri et al., 2013); Global Flood Forecasting Information System (GLOFFIS; https://www.globalfloodforecast.com/) and World-Wide HYPE (WWH; https://hypeweb.smhi.se/; Arheimer et al., 2020).

GloFAS is the global flood service of the European Commission's Copernicus Emergency Management Service (CEMS), an operational system for monitoring and forecasting floods across the world with over 6000 registered users in March 2021. The service and data are available through a free and open license and the system is designed to help decision makers and forecasters in sectors such as national and international water authorities, water resources managers, hydropower companies, civil protection authorities, and international humanitarian aid organisations. GloFAS is not designed to be a replacement for

local operational hydrological forecasting systems; in many parts of the world however a local or national system for operational forecasts of river discharge does not yet exist so it might be the only information available. GloFAS covers all river basins out to medium- and extended-range lead times (30-days ahead) and updated daily, with GloFAS-Seasonal (Emerton et al., 2018) updated monthly out to a 16-week lead time. Therefore, it has been used to complement local forecast systems by allowing forecasters to gain information on surrounding and upstream basins, monitoring for potential flood

signals where advanced warning is needed.

Given its global-scale, GloFAS can be used for providing daily assessments on potential upcoming flood events for the whole globe. This global overview is required by several users. For example, GloFAS is used daily as the main information source to monitor existing and upcoming river flood events and report back potential risks of flood impacts to the Emergency Response Coordination Centre (ERCC) of the European Commission, as part of the Aristotle-ENHSP project

(European Natural Hazard Scientific Partnership, http://aristotle.ingv.it/tiki-index.php, last accessed: 10 September 2020). Example real-world use cases of GloFAS include supporting the humanitarian response to the devastating floods that affected large parts of Mozambique, Malawi, and Zimbabwe in the wake of tropical cyclones Idai in March 2019 following a request from the Department for International Development of the UK government (Magnusson et al., 2019, Emerton et al., 2020) and during the 2020 monsoon season by the Bangladesh Flood Forecasting and Warning Centre (FFWC) (Hossain et

al., 2020).

GloFAS has been developed together by the Joint Research Centre (JRC) of the European Commission, the University of Reading, and the European Centre for Medium-Range Weather Forecasts (ECMWF), and was originally designed for large river basins and transboundary rivers. The system went pre-operational in July 2011 (Alfieri et al., 2013), becoming a fully operational 24/7 supported service in April 2018 (version 1.0, upgraded to Version 2.0 in November 2018). GloFAS version 2.1 was released on 5 November 2019 (GloFAS user wiki: https://confluence.ecmwf.int/display/COPSRV/GloFAS+v2.1, last accessed 14 September 2020).

There are two major gaps in data service delivery of the current generation of global hydrological forecasting, including GloFAS: Firstly, forecasts are generally issued as post-processed information (e.g. focusing on river discharge exceeding pre-defined flood thresholds) shown as maps and graphics on a dedicated web interface, but the raw data is not readily available to users. Having fast access to post-processed information has the advantage of providing an overview of the forecast output as an active flood event unfolds. However, not also having direct access to the raw data precludes the use in further downstream applications (e.g. impact modelling, multi-model forecast systems, production of value-added products for specific sectors such as river transport and hydropower industries, and advancement in techniques requiring large-scale datasets such as machine learning). Secondly, 'reforecasts' (i.e. forecasts for a set of past dates, also known as hindcasts) consistent as possible with the real-time forecasting system, ideally updated for each major model cycle upgrade, have not been made publicly available, limiting both global and user-specific local evaluation of forecast skill.

As part of the continued evolution of GloFAS in light of the aforementioned service gaps, the GloFAS real-time forecasts and a long-term and large-sample set of reforecasts will be made available to users as part of the forthcoming release of GloFAS version 2.2. This paper describes how the GloFAS forecast datasets (real-time and reforecasts) are generated, the methodology implemented for the forecast skill evaluation, and provides a global overview of the forecast skill assessment results that form the scientific basis for a new forecast skill layer on the GloFAS Web Map Viewer.

## 2 GloFAS components, configuration and data

The GloFAS hydrological forecasting system couples global numerical weather prediction (NWP) with hydrological modelling to produce ensemble forecasts of river discharge operationally each day across the world. The key model components of GloFAS version 2.1 (identical to forthcoming version 2.2 (expected launch in Autumn 2020), the latter being a minor service-only upgrade of with data access and new information layers added to the service) are shown in Figure 1 with a summary of real-time forecast (Sect. 2.1), reforecast (Sect. 2.2), and reanalysis (Sect.2.3) configurations in Table 1. Individual GloFAS model components have already been published in the scientific literature and hence are not described in detail here (Appendix A).

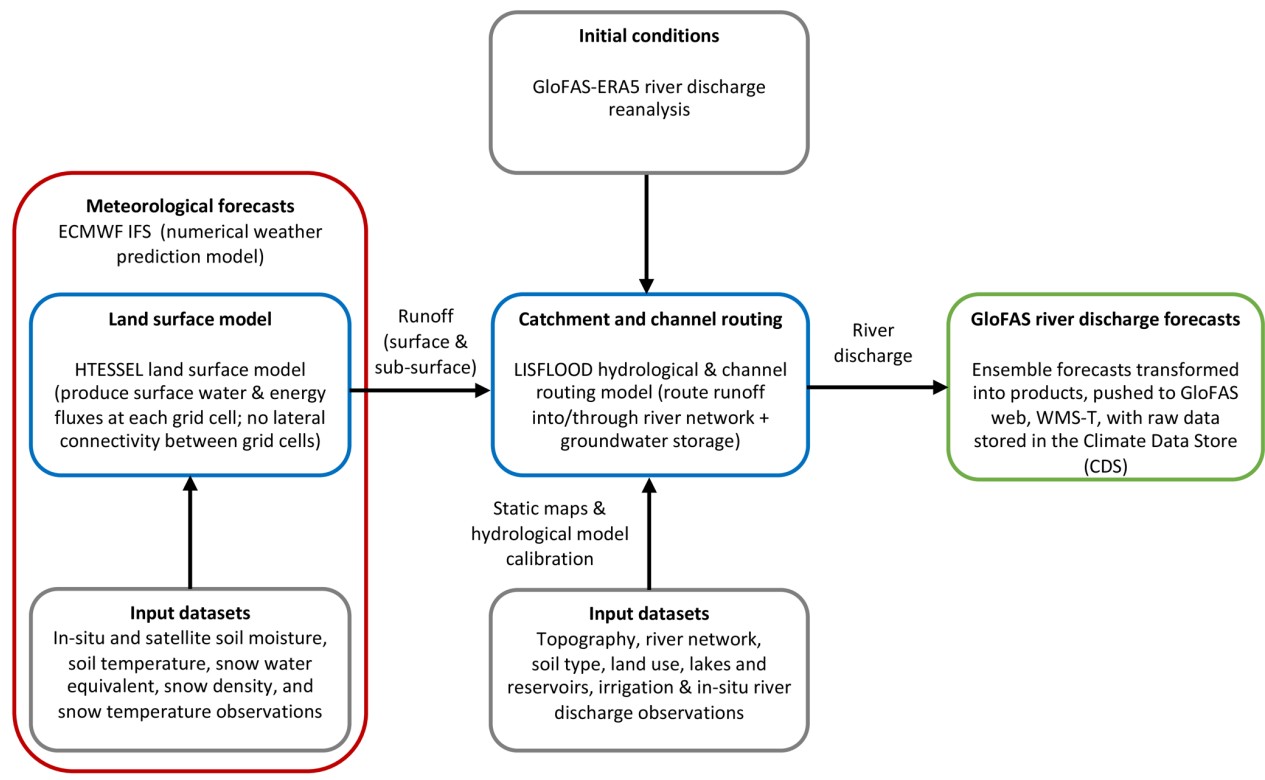

**Figure 1: Key components of GloFAS version 2.1/2.2.**

**Table 1: Real-time forecast and reforecast configurations for GloFAS version 2.1/2.2.**

| | GloFAS real-time forecasts | GloFAS reforecasts |
|---|---|---|
| GloFAS version[a] | 2.1/2.2 | 2.1/2.2 |
| ECMWF IFS version[b] (including HTESSEL) | 46r1 (5 November 2019 to 29 June 2020) <br> 47r1 (from 30 June 2020) | 45r1 (1 January to 10 June 2019) <br> 46r1 (11 June to 31 December 2019) |
| LISFLOOD version and calibration | Hirpa et al. (2018) | Hirpa et al. (2018) |
| Hydrological forecast initialisation | Latest GloFAS-ERA5 reanalysis with any temporal gap until real-time (i.e. 'fill up') with control (CTL) member of ENS | GloFAS-ERA5 |
| Variable | River discharge ($m^3 s^{-1}$) | River discharge ($m^3 s^{-1}$) |
| Time step | 24 h | 24 h |

| | | |
|---|---|---|
| Horizontal resolution | 0.1° (~ 11 km at equator) | 0.1° (~ 11 km at equator) |
| Lead time | 1 to 30 days | 1 to 30 days |
| Number of ensemble members | 51 | 11 |
| (Re)forecast frequency | Daily at 00 UTC | Twice per week (Mondays and Thursdays) at 00 UTC (104 start dates per year) |
| (Re)forecast period | 5 November 2019 to present | January 1999 to December 2018 |

[a] https://confluence.ecmwf.int/display/COPSRV/GloFAS+versioning+system
[b] https://www.ecmwf.int/en/forecasts/documentation-and-support/changes-ecmwf-model

**2.1 GloFAS real-time forecasts**

GloFAS is driven by the NWP model of the European Centre for Medium-Range Weather Forecasts (ECMWF), known as the Integrated Forecasting System (IFS). The current operational IFS model cycle is 47r1, implemented on 30 June 2020

(https://confluence.ecmwf.int/display/FCST/Implementation+of+IFS+Cycle+47r1, last accessed: 14 September 2020). Because the atmosphere is a chaotic system, ECMWF ensemble forecasts (ENS) are used to account for the inherent uncertainty and provide probabilistic forecasts in GloFAS (Figure 2). ECMWF ENS (~18km horizontal resolution) produces 51 ensemble members operationally out to a lead time of 15 days twice per day at 00:00 and 12:00 UTC. Ensemble members are comprised of a single 'control' (CTL) member which is generated from the most accurate estimate of current conditions

and the remaining 50 members which have their initial conditions perturbed to provide a range of possible future weather states. Twice per week (on Monday and Thursday) ECMWF ENS is extended to run to 46 days ahead at a coarser resolution (~36 km horizontal resolution), although in GloFAS only days 16 to 30 are used. The ECMWF ENS is run at a 6-hourly forecast time step and for ingestion into the GloFAS hydrological modelling chain, data from the 00 UTC run is extracted and aggregated to 24-houly time step.

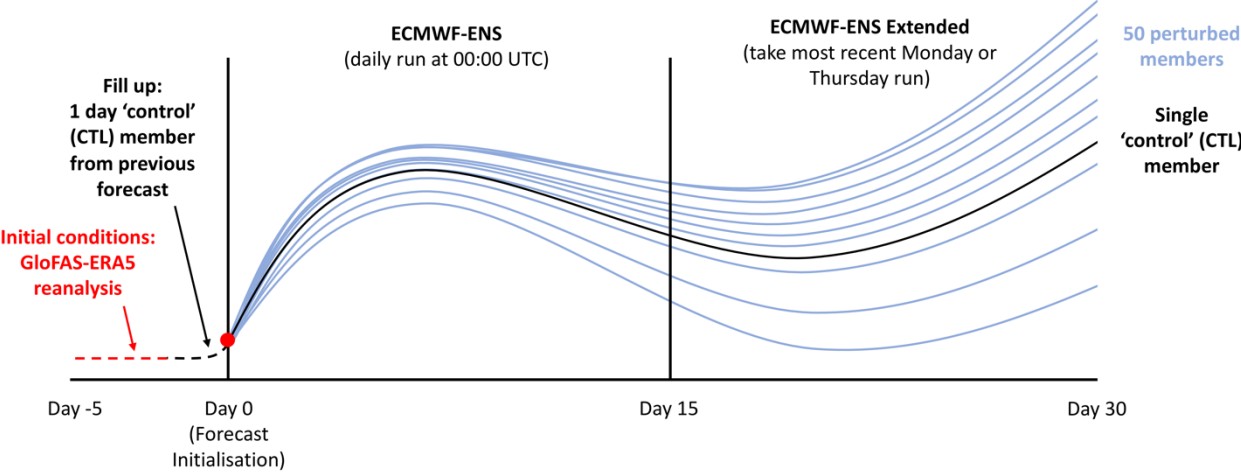

**GloFAS 30-day river discharge ensemble forecast**

**Figure 2: Schematic of a single GloFAS 30-day real-time river discharge ensemble forecast initialised at day 0 for GloFAS version 2.1/2.2.**

The hydrological modelling components of GloFAS (Figure 1) comprises the land surface model of ECMWF IFS, HTESSEL (Hydrology Tiled ECMWF Scheme for Surface Exchanges over Land; Balsamo et al., 2009), and LISFLOOD, a spatially distributed grid-based hydrological and channel routing model (van der Knijff et al., 2010). Precipitation is transformed to surface and sub-surface runoff in HTESSEL, with groundwater and channel routing processes simulated in LISFLOOD. In HTESSEL, excess precipitation and snowmelt are partitioned as surface runoff or infiltrated into a four-layer soil column (7 cm depth for top layer and then 21, 72, and 189 cm) at each IFS grid cell, before draining from the bottom of the soil column as sub-surface runoff.

Output from HTESSEL is downscaled to the GloFAS 0.1° (~11 km) gridded river network using the nearest neighbour method before being input to LISFLOOD. Surface runoff is then routed through the river network using the kinematic wave approach. Sub-surface runoff is used as input to the LISFLOOD groundwater module representing both base flow and faster groundwater pathways; it consists of two parallel linear reservoirs (upper zone for quick and lower zone for slower groundwater flow) that store and subsequently transport water to the river channel with a time delay. Groundwater and river routing parameters were calibrated against river discharge observations for 1287 catchments globally by Hirpa et al. (2018). A total of 463 of the largest lakes (surface area > 100 km$^2$) and 667 largest reservoirs have been incorporated into the GloFAS river network (Zajac et al., 2017). Reservoir outflow is calculated with a set of four rules depending on the current reservoir filling level (see Burek et al., 2013).

GloFAS real-time river discharge forecasts are produced operationally once per day using the ECMWF ENS initialised at 00:00 UTC (Figure 2). Initial hydrometeorological conditions are provided by the latest near-real-time GloFAS-ERA5 river discharge reanalysis (Harrigan et al. (2020a)  and Section 2.3), a product publicly available to users 2 to 5 days behind real time through the CDS. To fill this 2 to 5-day gap between the latest available GloFAS-ERA5 data and real-time initialisation

of the GloFAS forecast, the first 24 h period from the single ECMWF ENS CTL member from the preceding days forecast is used as 'fill up' (see Figure 2).

The final stage in the real-time forecast production is to generate plots and maps from the raw data highlighting possible upcoming flood events (see https://confluence.ecmwf.int/display/COPSRV/Overall+GloFAS+product+summary (last accessed: 14 September 2020) for a complete description of all GloFAS products. These products are pushed each day to the GloFAS Web Map Viewer and are freely available to users ( https://www.globalfloods.eu/ and available as Web Map Service with temporal requests each day, WMS-T: https://confluence.ecmwf.int/display/COPSRV/GloFAS+Web+Services,

last accessed: 14 September 2020). The raw real-time forecast GloFAS river discharge data, together with corresponding metadata, are then stored in the user-friendly data repository, the Copernicus Climate Data Store (CDS; https://cds.climate.copernicus.eu/#!/home, last accessed 10 September 2020) for use in downstream applications. Full details on data access can be found in Section 4.3.

## 2.2 GloFAS reforecasts

The quality of any forecast system can be evaluated by comparing a set of past forecasts with their corresponding observations (Jolliffe and Stephenson, 2012; Wilks, 2011). The set of past forecasts can be archived forecasts from the operational forecast system or a dedicated set of 'reforecasts' (also known as hindcasts) that are computed retrospectively using the same (or as close as possible) model as the real-time forecast for a number of past dates.

The set of past forecasts used to evaluate the skill of GloFAS (Sect. 3) were generated from the ECMWF 20-year operational
reforecasts. Compared with archived forecasts, using reforecasts has the advantage of being generated from the latest NWP configuration, generally more stable than archived forecasts produced from different model cycles. Typically, there is a new ECMWF IFS cycle release every 6 to 18 months (Table 1). In addition, changes can be made to the IFS or GloFAS modelling system components independently from the full ECMWF IFS cycle release (see https://confluence.ecmwf.int/display/COPSRV/GloFAS+versioning+system for a description of GloFAS release cycles since
its operational launch). The last 10 years of GloFAS archived forecasts contain at least 19 different ECMWF IFS model evolutions. Whilst not all IFS model changes impact the terrestrial water cycle, it is likely that there are significant changes in forecast errors between each model evolution, making the evaluation inconsistent through time. In contrast, the use of reforecasts has a number of advantages compared to using archived forecasts for forecast performance evaluation: 1) being run off-line, the latest hydrological routing component and simulation configuration (e.g. initial conditions) can be used,
providing a stable simulation of river discharge processes; 2) for the period 1 January 2019 to 31 December 2019, the IFS experienced only one upgrade (11 June 2019); and 3) a large-sample of ECMWF-ENS reforecasts are available (20 years long), albeit with a smaller size ensemble than the real-time simulation (11 members instead of 51), allowing for robust evaluation of forecast skill.

ECMWF uses an "on-the-fly" configuration to generate a continuous large reforecast sample, while balancing the
170 computational resources needed to run the operational global NWP. A reforecast task is run twice per week (on Mondays

and Thursdays) in parallel to the real-time forecast, using ERA5 atmospheric reanalysis (Hersbach et al., 2020) for initial conditions of past dates. A reforecast of the corresponding date for the previous 20 years is produced with a reduced number of 11 ensemble members but using the same model version as real-time (Vitart, 2014). For example, on Thursday 3 January 2019 a real time forecast based on IFS cycle 45r1 as well as a retrospective reforecast for 3 January for 20 years in the past (i.e. 3 January 1999 to 3 January 2018) was produced and archived. On Monday 7 January 2019, the process was repeated with reforecasts run for 7 January 1999 to 7 January 2018, and so on each Monday and Thursday operationally (Figure 3).

The GloFAS reforecast used here and made available was generated during the full calendar year of 2019 (i.e. Thursday 3 January to Monday 30 December). It is an ensemble containing forecast simulations of 104 start dates per year for the previous 20 years 1999 to 2018 (2080 start dates in total) composed of 11 members and running for lead times 1 to 30 days at a 24 h time step (Table 1). The river discharge reforecast was initialised from GloFAS-ERA5 (Sect. 2.3) and forced by ECMWF-ENS reforecast runoff from the twice weekly, 11-member, 20-year ECMWF meteorological ensemble reforecasts.

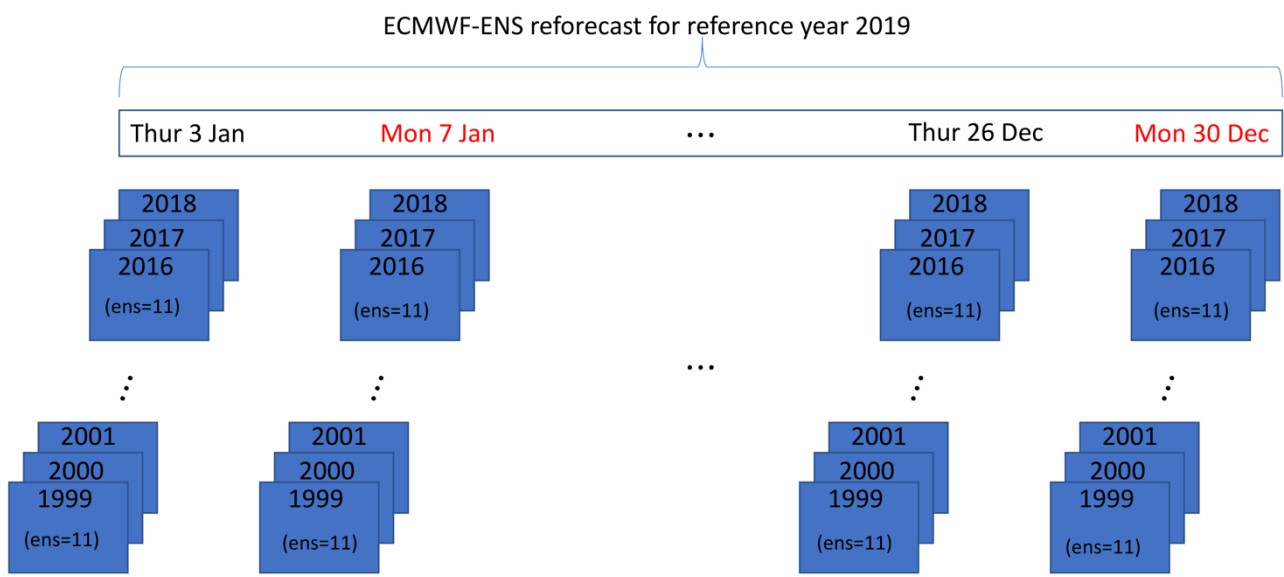

Figure 3: ECMWF-ENS reforecast schematic for the reference period January to December 2019.

## 2.3 GloFAS-ERA5 river discharge reanalysis

The GloFAS-ERA5 reanalysis dataset (Harrigan et al., 2020a) provides a spatio-temporally consistent estimate of daily historic river discharge. It is produced for every 0.1° river cell globally from 1979 to the present. It is updated operationally with a latency of 2 to 5 days behind real time, following the release of ERA5 atmospheric reanalysis (Hersbach et al., 2020). In GloFAS operational forecasts, GloFAS-ERA5 is used as initial conditions for the real-time forecasts (Figure 1 and Figure 2), and for calculating flood thresholds against which real time ensemble forecasts are compared to determine the probability

of a flood signal (Zsoter et al., 2020a). For the forecast evaluation undertaken here, GloFAS-ERA5 is used as initial conditions for reforecasts, to generate benchmark forecasts and as proxy observations to evaluate forecast skill.

The hydrological performance of GloFAS-ERA5 will have implications for the forecasts and reforecasts here. If for example GloFAS-ERA5 has poor hydrological skill in resolving hydrological dynamics, particularly the timing of river discharge, this would contribute to poorer forecasts. An evaluation of GloFAS-ERA5 against a global network of 1801 in-situ river discharge observation stations was undertaken by Harrigan et al. (2020a) and shown here in Figure 4 for context. They found the reanalysis is skilful in 86 % of catchments according to the modified Kling-Gupta Efficiency Skill Score against a mean flow benchmark (see Figure 4). The global median Pearson correlation coefficient is 0.61 with an interquartile range of 0.44 to 0.74. However, skill varies considerably with location with several regions such as central US, Africa, eastern Brazil, and western coast of South America having large systematic positive biases. For the evaluation presented here, GloFAS-ERA5 v2.1 data from 1979 to 2019 are used as downloaded from the Copernicus Climate Data Store (CDS): https://cds.climate.copernicus.eu/cdsapp#!/dataset/cems-glofas-historical?tab=overview (last accessed: 14 September 2020) (Harrigan et al., 2019).

## 3 Global forecast skill evaluation method

A first systematic evaluation of GloFAS hydrological forecast skill was carried using the operational version 2.1/2.2 at the global scale, across lead times from 1 to 30 days, based on the comprehensive set of 20-year reforecasts described in Section 2.2. The forecast evaluation methodology is set out below with the aim of being applied routinely to all future major releases of GloFAS, with the forecast skill statistics provided as a new forecast skill layer on the GloFAS Web Map Viewer as well as metadata information associated with the raw data provided on the Copernicus Climate Data Store (CDS). This aims to help users make better informed decisions on how, when and where GloFAS forecasts might be appropriate for their needs.

### 3.1 Data sample

There are 5.4 M GloFAS 0.1° river network cells covering the global land area, so to avoid excessive redundancy, forecast skill is calculated for a subset called the GloFAS diagnostic river points. There are 5997 of these diagnostic points in total used across the GloFAS project by both model developers and users for a range of purposes, such as displaying forecast hydrographs and associated detailed metadata at each point on the Web Map Viewer (known as GloFAS web points), diagnosing reanalysis and forecast errors, and tracking improvement between model upgrades. These river points drain catchment areas ranging from 1068 km$^2$ to 5,359,150 km$^2$ with a median area of 29,051 km$^2$ (Figure 4) and more information on each point can be found in Supplementary Information Table 1.

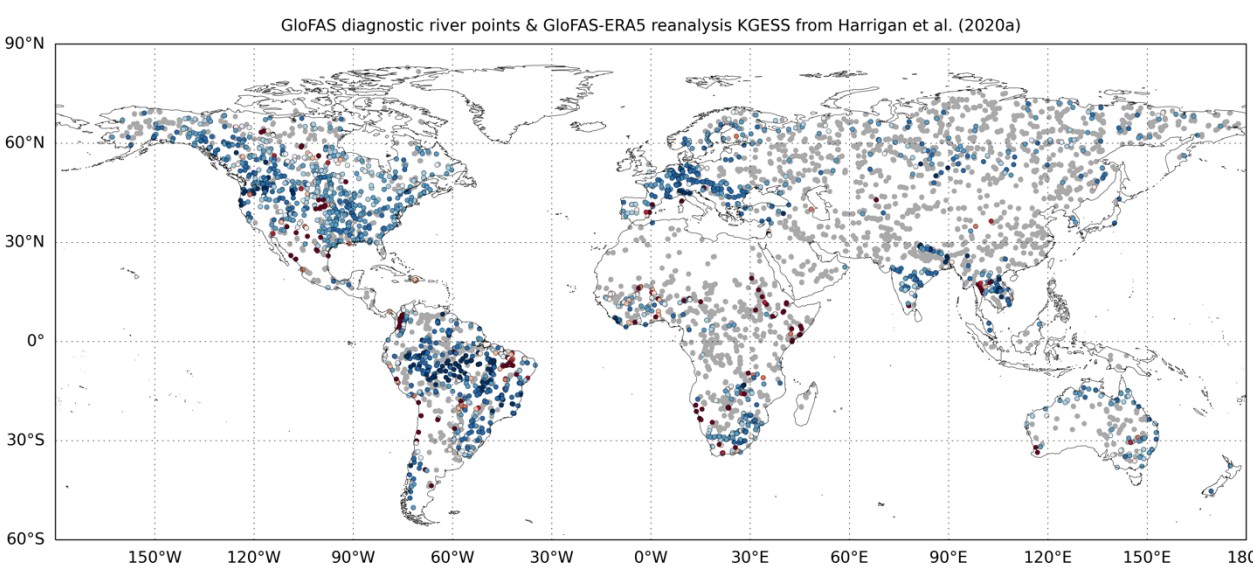

Figure 4. GloFAS diagnostic river points (n=5997) are highlighted by grey dots. Coloured dots show hydrological performance of GloFAS-ERA5 river discharge reanalysis against a subset of GloFAS diagnostic river points with observations (n=1801) from Harrigan et al. (2020a) using the modified Kling–Gupta efficiency skill score (KGESS). Optimum value of KGESS is 1. Blue (red) dots show catchments with positive (negative) hydrological skill.

## 3.2 Benchmark forecasts

Forecast skill refers to the relative accuracy of a set of forecasts with respect to a set of standard reference or *benchmark forecasts* (Wilks, 2013). When designing a forecast evaluation experiment, a critical consideration is the selection of a benchmark forecast that has sufficient skill discrimination, i.e. is not too simple and represents as closely as possible the observations (Pappenberger et al., 2015).

Following Pappenberger et al. (2015) and because GloFAS produces seamless forecasts across short-, medium- and extended-range lead times (day 1 to 30), two benchmarks are considered here, each calculated for all GloFAS diagnostic river points: *persistence*, typically used for short-range lead times where the forecast signal is dominated by serial correlation of river discharge, and *climatology*, typically used for longer lead times where the forecast signal is dominated by the seasonality of river discharge defined as follows:

- **persistence benchmark forecast** defined as the single GloFAS-ERA5 daily river discharge of the day preceding the reforecast start date. The same river discharge value is used for all lead times.
- **climatology benchmark forecast** based on a 40-year climatological sample (1979-2018) of moving 31-day windows of GloFAS-ERA5 river discharge reanalysis values, centred on the date being evaluated (+- 15 days). From each 1240-valued climatological sample (i.e. 40 years × 31-day window), 11 fixed quantiles (Qn) at 10 %

intervals were extracted (Q0, Q10, Q20, ⋯ , Q80, Q90, Q100). The fixed quantile climate distribution used therefore varies by lead time, capturing the temporal variability in local river discharge climatology.

## 3.3 Skill score

The ensemble forecast performance is evaluated using the Continuous Ranked Probability Score (CRPS) (Hersbach, 2000), one of the most widely used headline scores for probabilistic forecasts. The CRPS compares the continuous cumulative distribution of an ensemble forecast with the distribution of the observations. It has an optimum value of 0 and measures the error in the same units as the variable of interest (here river discharge in $m^3$ $s^{-1}$). It collapses to the Mean Absolute Error (MAE) for deterministic forecasts, which is important here as the persistence benchmark forecast we use is deterministic. The CRPS is expressed as a skill score, CRPSS, to calculate forecast skill which measures the improvement in GloFAS over the benchmark forecast and is given in Eq. (1):

$$CRPSS = 1 - \frac{CRPS_{fc}}{CRPS_{bench}}, \qquad (1)$$

where $CRPS_{fc}$ is the CRPS of the forecast against observations and $CRPS_{bench}$ is the CRPS of the benchmark forecast against observations. A CRPSS value of 1 indicates a perfect forecast, CRPSS > 0 shows forecasts are more skilful than the benchmark, CRPSS = 0 shows forecasts are only as accurate as the benchmark, and CRPSS < 0 means that forecasts are less skilful than the benchmark forecast.

The CRPSS was calculated using GloFAS reforecasts over 1999 to 2018 generated in Section 2.2 using both persistence and climatology benchmark forecasts (Sect. 3.2) and verified against GloFAS-ERA5 river discharge reanalysis used as proxy observations (following Alfieri et al., 2014) at each of the 5997 GloFAS diagnostic river points. Calculating forecast skill against proxy observations such as reanalysis is common in hydrological forecasting as it has the advantage of providing a spatio-temporally complete picture of forecast skill, currently not possible based on availability of the current global in situ observed river network (Lavers et al., 2019). It also allows the forecast predictability range to be isolated in the absence of systematic hydrological model errors. There is a disadvantage of forecast evaluation against proxy observations for catchments that represent hydrological dynamics poorly. While Harrigan et al. (2020a) demonstrate the performance of GloFAS-ERA5 reanalysis is largely hydrologically skilful, readers should be aware that there are areas where performance is poor and that there are large parts of the world where the performance is unknown due to the lack of in situ observations to evaluate against (Figure 4).

## 4 Results and discussion

 ### 4.1 GloFAS forecast skill

### 4.1.1 GloFAS skill by lead time

Overall, GloFAS version 2.1/2.2 is skilful at the global scale for the majority of catchments across all lead times analysed (Figure 5a). The CRPSS against persistence decays exponentially as a function of lead time out to around 15 days lead time, then stabilises reaching a minimum of 0.48 by day 22. The CRPSS at day 1 is 0.96 (interquartile range of 0.88 to 0.99), day 3
= 0.82 (0.62, 0.94), day 5 = 0.70 (0.50, 0.87), day 10 = 0.55 (0.37, 0.72). The CRPSS against climatology begins higher than persistence and decays continuously towards day 30. At day 15 is 0.47 (0.27, 0.68), day 20 = 0.37 (0.17, 0.58), day 25 = 0.30 (0.11, 0.49), day 30 = 0.25 (0.06, 0.43).

As the global median CRPSS against climatology becomes lower than against persistence (0.49 versus 0.50, respectively) from day 14, we present and discuss all forecast skill from short- (1- to 3-day) to medium-range (5- to 10-day) lead times
275 calculated against the persistence benchmark forecast, and from extended lead times (15- to 30-days) calculated against the climatology benchmark forecast. The aid the readers interpretation of the CRPSS, individual CRPS components used in equation 1 (i.e. $CRPS_{fc\_GloFAS}$, $CRPS_{bench\_persistence}$, and $CRPS_{bench\_climatology}$) are also shown in Figure 5b expressed as a global median across all lead times.

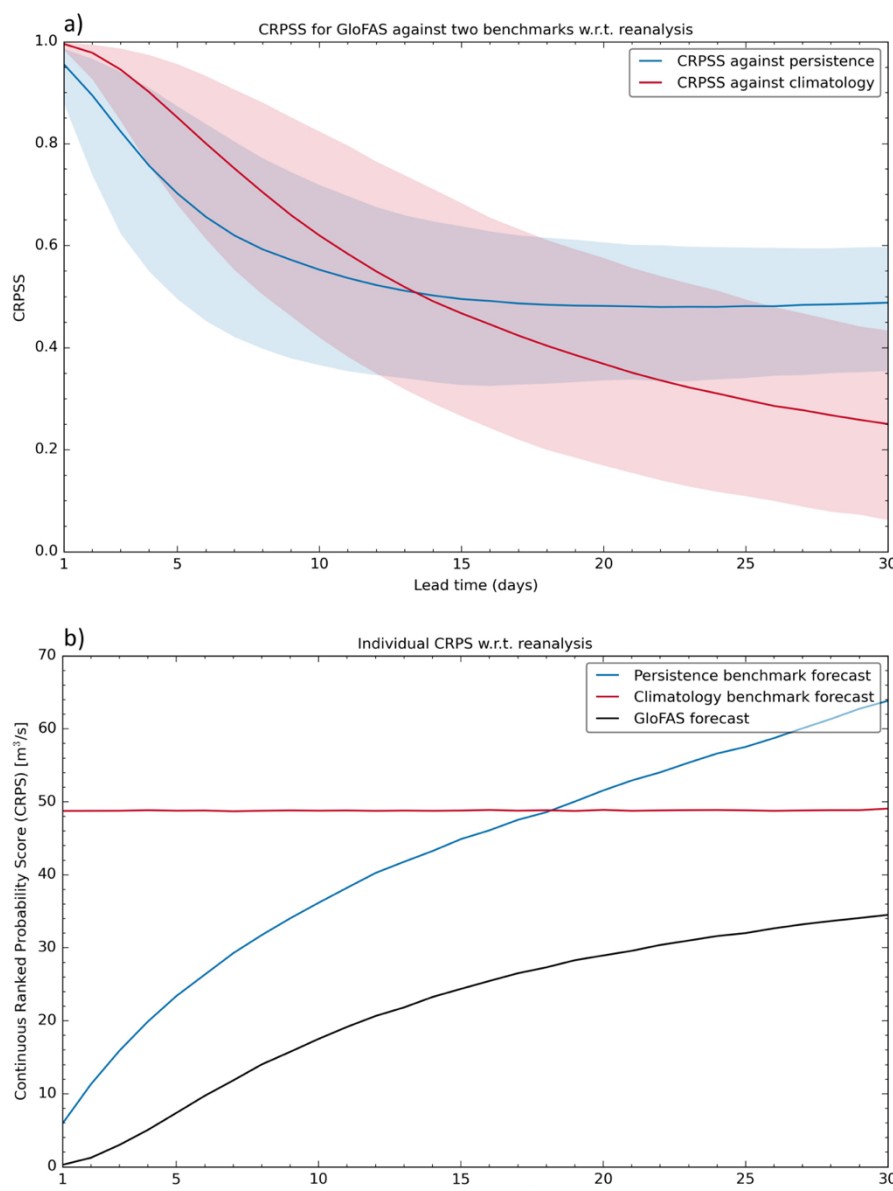

Figure 5. Skill of GloFAS 2.1/2.2 with global median Continuous Ranked Probability Skill Score (CRPSS) for reforecasts against persistence (red line) and climatology (blue line) benchmarks from 1- to 30-day lead times with respect to GloFAS-ERA5 river discharge reanalysis at 5997 diagnostic river points (a). The interquartile range of CRPSS values at each lead time are shown by semi-transparent bands. Corresponding individual CRPS components used in equation 1 for GloFAS forecasts (black line), persistence benchmark (blue line) and climatology (red line) shown as a median across at the diagnostic river points (b).

### 4.2.2 Spatial distribution of GloFAS skill

At short-range lead times (1 and 3 days), GloFAS is skilful compared to the persistence benchmark forecast in over 96 % of catchments (Figure 6). In the medium-range at day 5, GloFAS remains skilful for 93 % of catchments. Regions with the highest skill (CRPSS >= 0.8) include South America, especially the Amazon basin, the US, southern Africa, central Asia,

and eastern Australia. There are notable clusters of catchments with negative skill (i.e. CRPSS < 0) mainly located in northern polar latitudes above 60° N as well as in the Congo River Basin. The global median CRPSS at day 5 for catchments located in the northern polar climate region is 0.63 compared to 0.70 and 0.73 for extratropics and tropics, respectively (Figure 8a). By day 10, the strength of skill has decreased, but 89 % of catchments remain skilful (i.e. CRPSS > 0).

For extended-range lead times shown in Figure 7, GloFAS is skilful compared to the climatology benchmark forecast in 89 % of catchments by day 15, reducing to 86 %, 83 % and 80 % by lead times 20-, 25- and 30-days, respectively. The regions of highest skill are similar to those for short- and medium-range, with areas of negative skill expanding to tropical Africa, a large region in central and northern Asia, and western coast of South America. The global median CRPSS at day 20 for catchments located in the broader tropics (latitudes 23° S to 23° N) is 0.31 compared to 0.40 and 0.37 for the extratropics and polar climate regions, respectively (Figure 8a).

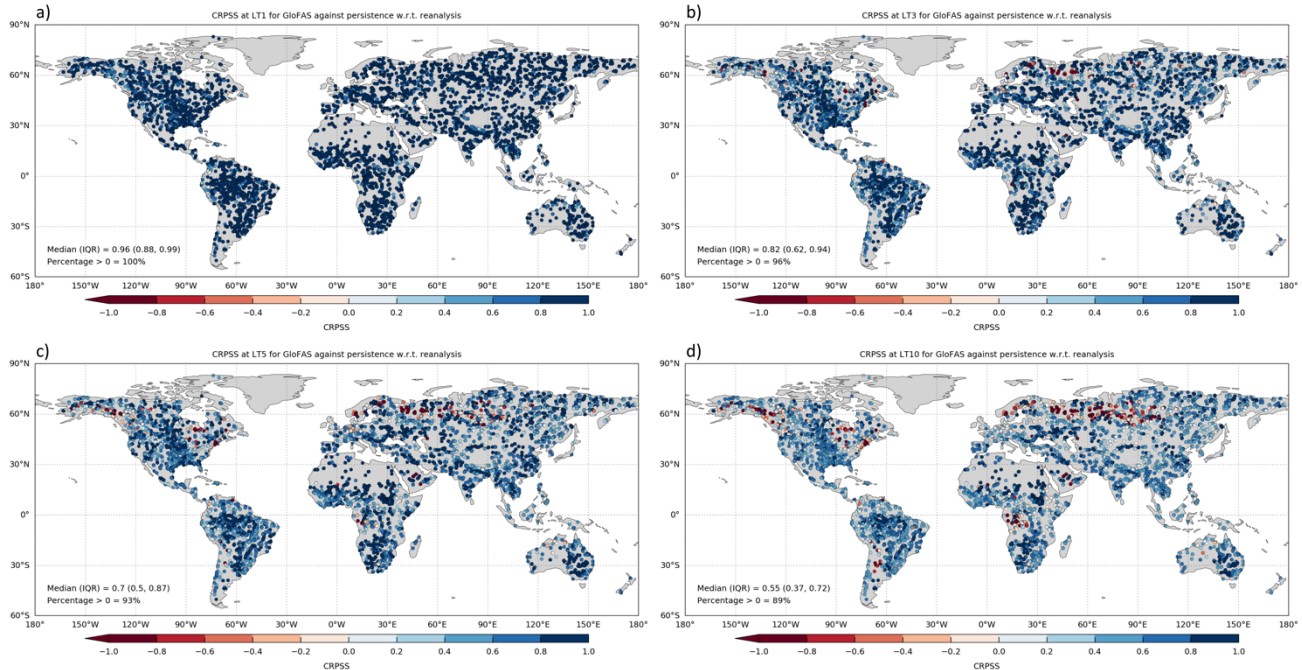

Figure 6: GloFAS 2.1/2.2 with Continuous Ranked Probability Skill Score (CRPSS) for reforecasts against the persistence benchmark for short- to medium-range lead times (1- (a), 3- (b), 5- (c), and 10-days (d)) with respect to GloFAS-ERA5 river discharge reanalysis at 5997 diagnostic river points. Optimum value of CRPSS is 1. Blue (red) dots show catchments with positive (negative) skill.

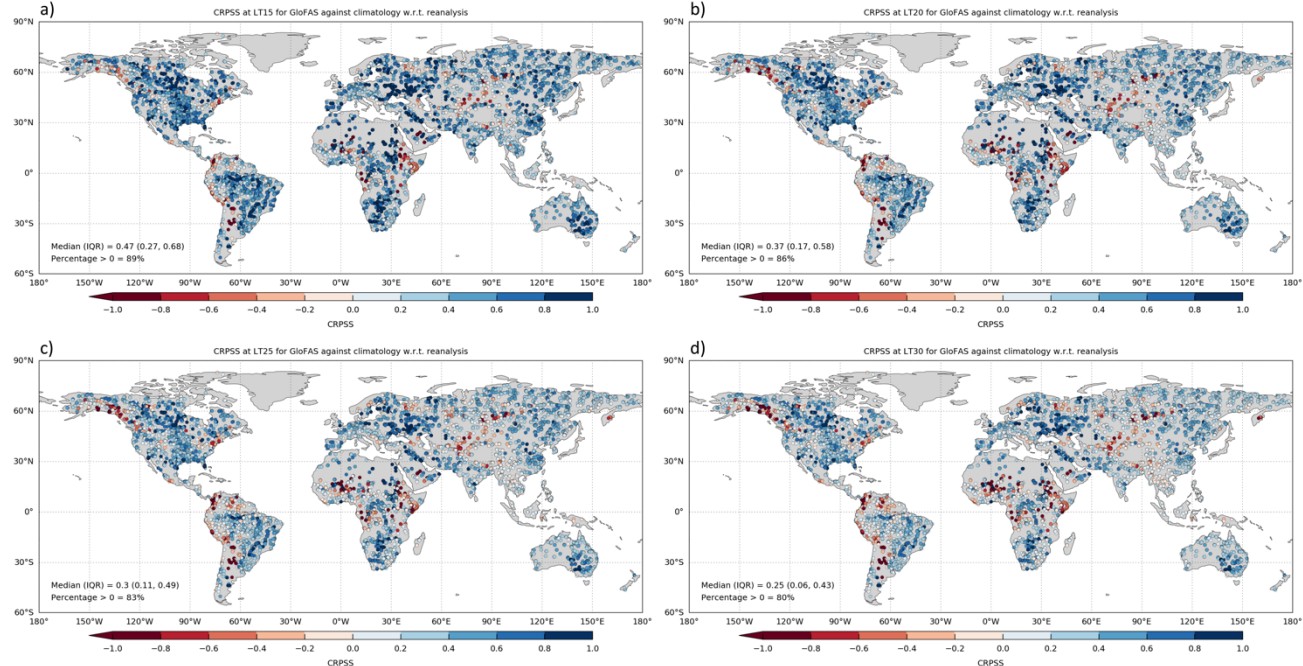

Figure 7: GloFAS 2.1/2.2 Continuous Ranked Probability Skill Score (CRPSS) for reforecasts against the climatology benchmark for extended lead times (15- (a), 20- (b), 25- (c), and 30-days (d)) with respect to GloFAS-ERA5 river discharge reanalysis at 5997 diagnostic river points. Optimum value of CRPSS is 1. Blue (red) dots show catchments with positive (negative) skill.

### 4.2.3 GloFAS skill by catchment area and hydrological flashiness

GloFAS skill using CRPSS for representative medium-range (using 5-day) and extended-range (using 20-day) lead times is correlated against catchment area in Figure 8b and the Richards-Baker Flashiness Index (RB Index; Baker et al., 2004) in Figure 8c using the Spearman rank correlation coefficient (Rho). Forecast skill is moderately positively correlated with catchment area (Rho = 0.50 (0.31) for 5- (20-) day lead times); catchments with larger areas have higher skill. This is consistent with findings in Ireland (Donegan et al., 2020; Quinn et al., 2021), and at the European scale from EFAS (Alfieri et al., 2014). While catchments with no skill (CRPSS ≤ 0) tend to be smaller, the majority of catchments with areas ranging between 1000 to 10,000 km$^2$ are skilful.

The RB Index is calculated by dividing the pathlength of day-to-day river discharge changes by total river discharge for a given time interval. For each catchment, the RB Index was extracted from GloFAS-ERA5 over the time interval 1979 to 2019. The index provides a useful summary of hydrological functioning of a catchment. Catchments with a high RB Index tend to have flashy hydrological response and are characterised as smaller upland catchments with increased frequency and magnitude of storm events, whereas catchments with a low RB Index tend to be slower responding larger catchments with higher baseflow components (Baker et al., 2004). Forecast skill is weakly to moderately negatively correlated with RB Index (Rho = -0.21 (-0.40) for 5- (20-) day lead times); catchments with higher hydrological flashiness have lower skill. The link between higher catchment responsiveness and lower forecast skill has also been found in Ireland (Donegan et al., 2020;

Quinn et al., 2021), UK (Harrigan et al., 2018), Sweden (Girons Lopez et al., 2021), and at the European scale from EFAS (Pappenberger et al., 2015).

Formal attribution of the drivers of high and low hydrological forecast skill is outside the scope of this study but results point to several areas to prioritise research and development into model improvements. First, improving GloFAS forecast performance in smaller catchments with more flashy hydrological response should be a priority. This finding is expected given the relatively coarse horizontal (~11 km) and time (daily) resolution of a global-scale system such as GloFAS. A "hyperresolution" target in the order 1 km globally is required for hydrological prediction to be useful at local scales (Wood et al., 2011), but will bring computational, data and hydrological science challenges (Harrigan et al., 2020b). Second, hydrological forecast skill is inherently dependent on global NWP model skill. Prediction of convective rainfall, dominant in the tropics, remains a challenge in the current generation of NWP, including the ECMWF IFS (~18km horizontal resolution for ENS) used to force GloFAS forecasts (Haiden et al., 2021; Lavers et al., 2021). Progress is however already underway. Recent increases in supercomputer power has allowed ground-breaking kilometre-scale NWP to be tested with promising results showing that deep convection can be explicitly simulated rather than parameterised as it is currently, thus providing better representation of convective storm activity (Wedi et al., 2020). Assessing the hydrological impact of any new precipitation improvement needs to be prioritised. Thirdly, hydrological prediction in regions with more challenging hydroclimate conditions needs further investigation, particularly snowy and icy catchments in polar regions where simplified snow accumulation and melt processes as well as rain-on-snow events are known to be highly sensitive to error (Fehlmann et al., 2019). From this first order assessment GloFAS forecast performance can drop considerably for many catchments in these regions. Therefore, more work is needed to investigate how existing and new representations of snow processes can deliver more skilful river discharge forecasts.

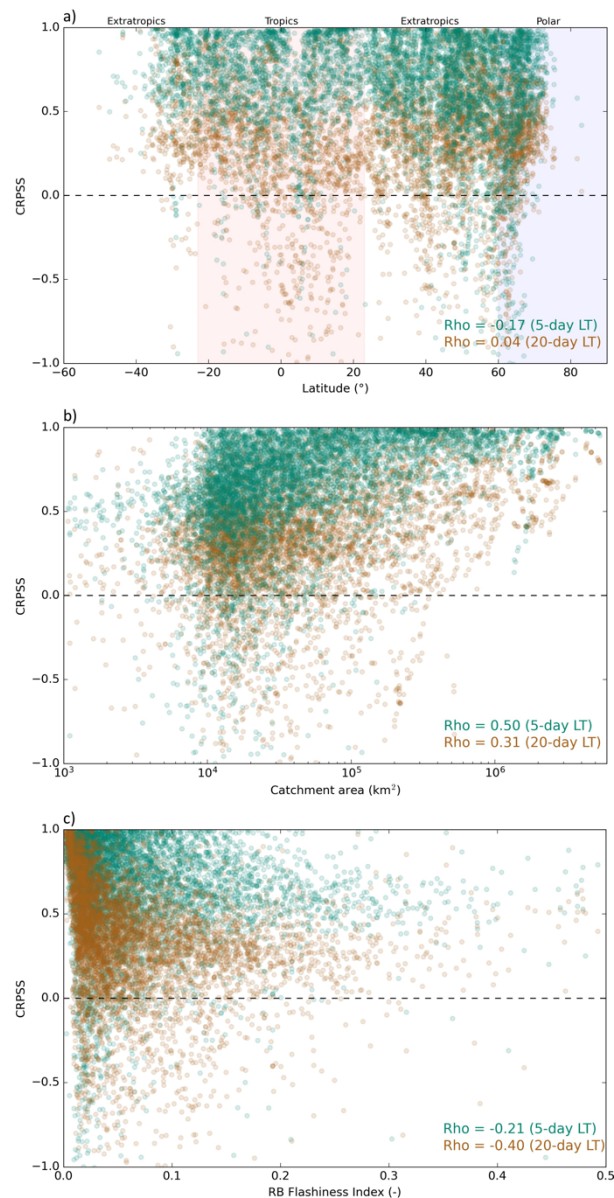

**Figure 8: GloFAS 2.1/2.2 Continuous Ranked Probability Skill Score (CRPSS) for 5-day (green dots; against persistence benchmark) and 20-day (brown dots; against climatology benchmark) lead times at 5997 diagnostic river points by degree latitude of the river point (a), catchment area (b), and RB Flashiness index (c). Spearman Rank correlation coefficients (Rho) for each combination given in text in the bottom right.**

## 4.2 New GloFAS headline forecast skill layer on Web Map Viewer

To help the interpretation and understanding of the quality of GloFAS 30-day forecasts, the forecast skill scores produced in this paper are be presented as new layer on the GloFAS Web Map Viewer since the release of GloFAS version 2.2 on 9 December 2020. Figure 9 shows a screenshot of the "Forecast skill" layer on the website. The new headline forecast skill

score is defined as the maximum lead time (in days), up to 30-days ahead, in which the CRPSS is greater than a value of 0.5, when compared to a persistence or climatology benchmark forecast using GloFAS-ERA5 as proxy observations. A threshold of CRPSS = 0.5 is chosen for the summary layer to distinguish the lead time in which a station is 'highly skilful' and is interpreted practically as the threshold at which the GloFAS forecast are 50 % more accurate than the respective benchmark forecast. The headline score is shown for the GloFAS reporting points. An example of the detailed skill information available for individual stations is shown for the Rhine at Lobith (Netherlands; G0337) in the inset of Figure 9. The headline score for this station is at day 7, when the CRPSS against persistence drops below the 0.5 threshold. When the station is clicked on the web interface, a 'pop out' window appears and includes two plots, the CRPSS across the 30-day lead time and corresponding individual CRPS components. This will provide vital information for forecasters when conducting forecasting assessment during emergency situations.

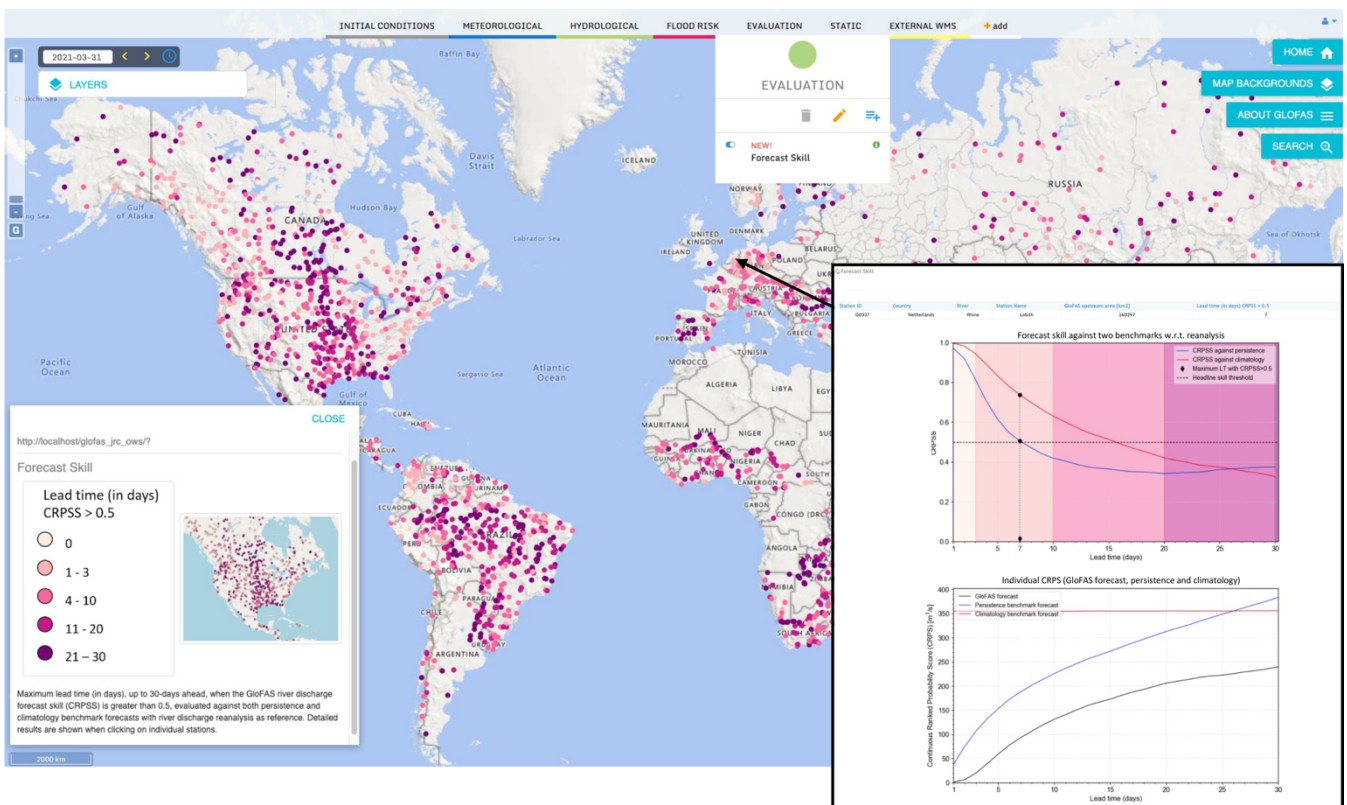

**Figure 9: GloFAS 30-day forecast skill layer for the headline score available on the GloFAS Web Map Viewer. The headline score is the maximum lead time (in days) the Continuous Ranked Probability Skill Scores (CRPSS) is greater than a value of 0.5, evaluated against a persistence or climatology benchmark forecast. Clicking on each GloFAS reporting point, a 'pop out' window shows the detailed CRPSS and CRPS across all lead times. An example for the Rhine at Lobith (Netherlands, G0337) is shown in the inset.**

## 4.3 Operational delivery of GloFAS data and metadata

The GloFAS global river discharge forecasts (real-time and reforecasts) and associated skill assessment analysis are provided free and openly by the European Commission Copernicus Emergency Management Service (CEMS). It follows the Copernicus open data policy that users shall have free, full, and open access to Copernicus Service Information. Users should however adhere to its terms and conditions available at https://www.globalfloods.eu/terms-of-service/.

The Copernicus Climate Change Service (C3S) Climate Data Store (CDS; https://cds.climate.copernicus.eu/cdsapp#!/home) hosts numerous global and regional reanalysis and forecast products, generally in the form of gridded records for Essential Climate Variables (ECVs), including river discharge data as a key terrestrial ECV. The CDS requires standardisation of data and metadata so that datasets are more useable and discoverable through the CDS metadata pages. Its website provides easy access to data through user-friendly download forms, as well as a CDS Python Application Programming Interface (API) to 380 allow programmatic access to data. An innovative feature of the CDS is its 'Toolbox', which makes it easier to handle large volumes of data by allowing users to make custom applications, filter data by geographical region and date range, and finally present the data using maps and charts directly through the CDS cloud infrastructure.

The GloFAS real-time river discharge forecasts from 5 November 2019 until present are available on the CDS and updated operationally every day: https://cds.climate.copernicus.eu/cdsapp#!/dataset/cems-glofas-forecast?tab=overview (last 385 accessed: 14 September 2020) (Zsoter et al., 2019). The GloFAS river discharge reforecasts for the period 1999 to 2018 are also available on the CDS and update ahead of each major model cycle release (currently version 2.2): https://cds.climate.copernicus.eu/cdsapp#!/dataset/cems-glofas-reforecast?tab=overview (last accessed: 9 December 2020) (Zsoter et al., 2020b). The CDS landing page for the GloFAS forecast dataset is shown in Figure 10. The forecast data are available in two ways. First, through the 'Data Download' tab whereby users can manually select options in a form for which 390 data they would like to download in either GRIB or NetCDF file format. Second, data can be retrieved through the dedicated Python CDS API; an example API retrieval script is shown in Appendix B for the forecast start date of 1 January 2020 for both the single control (CTL) forecast and 50 ensemble perturbed members out to a lead time of 30-days at 24 h steps and downloaded in NetCDF format. Note that users must register for a CDS account (for free) before gaining access. This landing page always provide access to the latest operational system, with the possibility to go through earlier versions in the 395 archive when searching through past dates. For users interested in the raw forecast skill scores calculated in this paper, they are provided for all GloFAS diagnostic river points through the 'Documentation' tab on the CDS as well as in Supplementary Table 1. See Figure 11 for an extract of the skill score information provided.

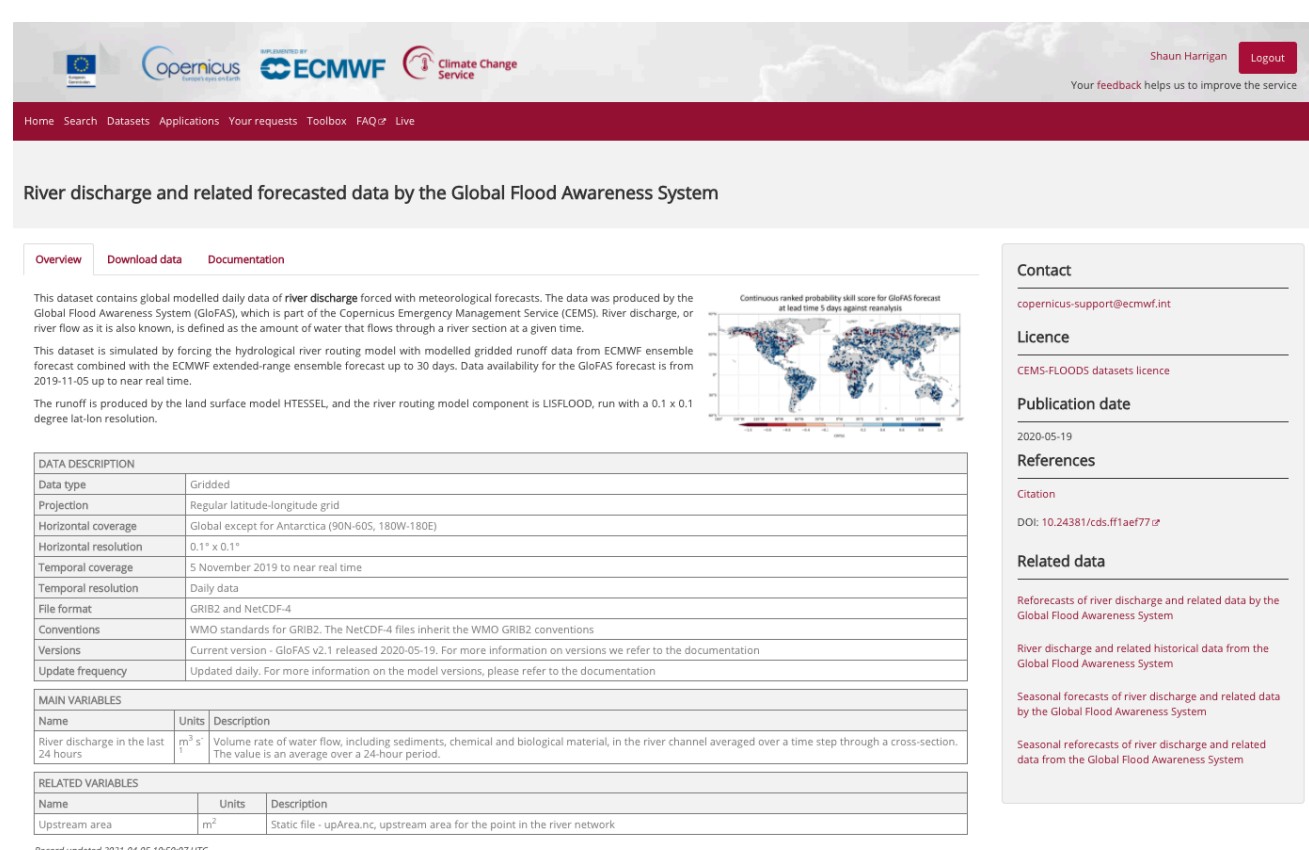

**Figure 10: The GloFAS river discharge forecast landing page on the C3SClimate Data Store (CDS: https://cds.climate.copernicus.eu/cdsapp#!/dataset/cems-glofas-forecast?tab=overview).**

| Country | GloFAS ID | Station filter | LT1 | LT2 | LT3 | LT4 | LT5 | LT6 | LT7 | LT8 | LT9 | LT10 | LT12 | LT14 | LT16 | LT18 | LT20 | LT25 | L30 |
|---|---|---|---|---|---|---|---|---|---|---|---|---|---|---|---|---|---|---|---|
| France | G1589 | 1 | 0.98 | 0.90 | 0.79 | 0.70 | 0.63 | 0.58 | 0.54 | 0.52 | 0.49 | 0.47 | 0.44 | 0.42 | 0.41 | 0.42 | 0.43 | 0.47 | 0.48 |
| France | G1590 | 1 | 0.83 | 0.74 | 0.69 | 0.65 | 0.63 | 0.59 | 0.55 | 0.51 | 0.48 | 0.45 | 0.43 | 0.40 | 0.36 | 0.33 | 0.31 | 0.30 | 0.31 |
| France | G1591 | 1 | 0.74 | 0.58 | 0.53 | 0.50 | 0.48 | 0.47 | 0.44 | 0.42 | 0.39 | 0.38 | 0.36 | 0.35 | 0.35 | 0.34 | 0.33 | 0.34 | 0.35 |
| France | G1592 | 1 | 0.95 | 0.75 | 0.31 | -0.11 | -0.35 | -0.45 | -0.49 | -0.49 | -0.51 | -0.50 | -0.46 | -0.42 | -0.37 | -0.19 | -0.03 | 0.11 | 0.16 |
| France | G1593 | 1 | 0.91 | 0.72 | 0.50 | 0.30 | 0.17 | 0.10 | 0.05 | 0.01 | -0.03 | -0.04 | -0.04 | -0.03 | 0.00 | 0.05 | 0.10 | 0.21 | 0.27 |
| France | G1594 | 1 | 0.73 | 0.64 | 0.57 | 0.52 | 0.49 | 0.45 | 0.41 | 0.36 | 0.34 | 0.31 | 0.29 | 0.27 | 0.25 | 0.19 | 0.16 | 0.15 | 0.17 |
| France | G1595 | 1 | 0.91 | 0.87 | 0.82 | 0.77 | 0.71 | 0.65 | 0.55 | 0.47 | 0.40 | 0.36 | 0.32 | 0.30 | 0.29 | 0.29 | 0.31 | 0.38 | 0.41 |
| France | G1596 | 1 | 0.94 | 0.91 | 0.88 | 0.82 | 0.73 | 0.63 | 0.50 | 0.39 | 0.31 | 0.27 | 0.23 | 0.22 | 0.22 | 0.24 | 0.27 | 0.37 | 0.41 |
| France | G1597 | 1 | 0.99 | 0.98 | 0.95 | 0.86 | 0.72 | 0.51 | 0.30 | 0.15 | 0.05 | 0.00 | -0.06 | -0.06 | -0.05 | -0.01 | 0.06 | 0.22 | 0.29 |
| France | G1598 | 1 | 0.86 | 0.76 | 0.69 | 0.62 | 0.58 | 0.56 | 0.53 | 0.49 | 0.47 | 0.46 | 0.42 | 0.40 | 0.36 | 0.33 | 0.31 | 0.29 | 0.29 |
| France | G1599 | 1 | 0.89 | 0.79 | 0.73 | 0.70 | 0.66 | 0.64 | 0.61 | 0.59 | 0.56 | 0.54 | 0.49 | 0.47 | 0.43 | 0.43 | 0.42 | 0.46 | 0.46 |
| France | G1600 | 1 | 0.99 | 0.96 | 0.92 | 0.86 | 0.81 | 0.75 | 0.70 | 0.66 | 0.63 | 0.60 | 0.55 | 0.50 | 0.48 | 0.45 | 0.44 | 0.44 | 0.44 |
| France | G1601 | 1 | 0.97 | 0.90 | 0.78 | 0.68 | 0.61 | 0.57 | 0.54 | 0.52 | 0.51 | 0.50 | 0.48 | 0.46 | 0.46 | 0.45 | 0.45 | 0.47 | 0.47 |
| France | G3694 | 1 | 0.79 | 0.71 | 0.66 | 0.60 | 0.57 | 0.56 | 0.53 | 0.52 | 0.50 | 0.51 | 0.47 | 0.44 | 0.42 | 0.40 | 0.40 | 0.42 | 0.42 |
| France | G3695 | 1 | 0.93 | 0.91 | 0.87 | 0.82 | 0.74 | 0.64 | 0.52 | 0.41 | 0.34 | 0.29 | 0.25 | 0.23 | 0.23 | 0.25 | 0.27 | 0.38 | 0.42 |
| France | G3699 | 1 | 0.75 | 0.62 | 0.49 | 0.41 | 0.37 | 0.37 | 0.36 | 0.35 | 0.35 | 0.35 | 0.33 | 0.34 | 0.34 | 0.35 | 0.35 | 0.38 | 0.39 |
| France | G3716 | 1 | 0.98 | 0.94 | 0.90 | 0.86 | 0.82 | 0.78 | 0.76 | 0.73 | 0.70 | 0.68 | 0.64 | 0.60 | 0.57 | 0.55 | 0.54 | 0.53 | 0.53 |
| France | G3851 | 1 | 0.86 | 0.78 | 0.67 | 0.63 | 0.61 | 0.57 | 0.55 | 0.52 | 0.52 | 0.50 | 0.47 | 0.45 | 0.44 | 0.43 | 0.44 | 0.45 | 0.47 |
| Germany | G0292 | 1 | 0.97 | 0.93 | 0.84 | 0.73 | 0.59 | 0.45 | 0.35 | 0.28 | 0.22 | 0.19 | 0.16 | 0.17 | 0.19 | 0.22 | 0.25 | 0.32 | 0.38 |
| Germany | G0297 | 1 | 0.98 | 0.91 | 0.82 | 0.69 | 0.54 | 0.41 | 0.31 | 0.24 | 0.19 | 0.16 | 0.15 | 0.16 | 0.18 | 0.22 | 0.25 | 0.32 | 0.38 |
| Germany | G0298 | 1 | 0.93 | 0.72 | 0.38 | 0.19 | 0.11 | 0.09 | 0.10 | 0.13 | 0.16 | 0.19 | 0.24 | 0.29 | 0.33 | 0.35 | 0.37 | 0.44 | 0.49 |
| Germany | G0301 | 1 | 0.99 | 0.93 | 0.74 | 0.49 | 0.30 | 0.17 | 0.09 | 0.03 | 0.01 | 0.00 | 0.02 | 0.06 | 0.10 | 0.13 | 0.17 | 0.24 | 0.30 |
| Germany | G0304 | 1 | 0.91 | 0.64 | 0.25 | 0.05 | -0.02 | -0.03 | -0.01 | 0.02 | 0.05 | 0.08 | 0.15 | 0.22 | 0.27 | 0.31 | 0.35 | 0.43 | 0.49 |
| Germany | G0306 | 1 | 0.98 | 0.97 | 0.95 | 0.92 | 0.86 | 0.79 | 0.72 | 0.65 | 0.59 | 0.53 | 0.45 | 0.39 | 0.35 | 0.33 | 0.32 | 0.32 | 0.34 |
| Germany | G0309 | 1 | 0.97 | 0.95 | 0.91 | 0.86 | 0.81 | 0.77 | 0.73 | 0.69 | 0.66 | 0.64 | 0.59 | 0.54 | 0.50 | 0.48 | 0.47 | 0.46 | 0.46 |
| Germany | G0310 | 1 | 0.97 | 0.95 | 0.91 | 0.86 | 0.81 | 0.77 | 0.73 | 0.69 | 0.66 | 0.64 | 0.59 | 0.54 | 0.50 | 0.48 | 0.47 | 0.46 | 0.46 |
| Germany | G0311 | 1 | 0.87 | 0.54 | 0.15 | -0.02 | -0.07 | -0.07 | -0.05 | -0.02 | 0.02 | 0.05 | 0.13 | 0.19 | 0.24 | 0.28 | 0.32 | 0.40 | 0.46 |
| Germany | G0313 | 1 | 0.98 | 0.94 | 0.86 | 0.71 | 0.55 | 0.41 | 0.30 | 0.23 | 0.18 | 0.15 | 0.14 | 0.15 | 0.18 | 0.21 | 0.24 | 0.32 | 0.38 |
| Germany | G0316 | 1 | 0.94 | 0.87 | 0.81 | 0.76 | 0.72 | 0.69 | 0.66 | 0.63 | 0.59 | 0.55 | 0.50 | 0.46 | 0.44 | 0.44 | 0.44 | 0.44 | 0.44 |
| Germany | G0320 | 1 | 0.86 | 0.51 | 0.13 | -0.03 | -0.09 | -0.08 | -0.06 | -0.03 | 0.01 | 0.04 | 0.12 | 0.19 | 0.24 | 0.28 | 0.31 | 0.40 | 0.46 |
| Germany | G0321 | 1 | 0.84 | 0.47 | 0.07 | -0.09 | -0.14 | -0.13 | -0.11 | -0.07 | -0.03 | 0.00 | 0.08 | 0.16 | 0.21 | 0.25 | 0.28 | 0.38 | 0.44 |
| Germany | G0323 | 1 | 0.83 | 0.40 | 0.07 | -0.08 | -0.17 | -0.20 | -0.19 | -0.16 | -0.15 | -0.12 | -0.06 | -0.01 | 0.04 | 0.08 | 0.12 | 0.22 | 0.30 |
| Germany | G0325 | 1 | 0.99 | 0.95 | 0.85 | 0.69 | 0.52 | 0.38 | 0.28 | 0.21 | 0.16 | 0.14 | 0.13 | 0.15 | 0.17 | 0.20 | 0.24 | 0.32 | 0.38 |
| Germany | G0329 | 1 | 0.99 | 0.95 | 0.85 | 0.69 | 0.52 | 0.38 | 0.28 | 0.21 | 0.16 | 0.14 | 0.13 | 0.15 | 0.17 | 0.20 | 0.24 | 0.32 | 0.38 |
| Germany | G0330 | 1 | 0.75 | 0.23 | -0.10 | -0.22 | -0.23 | -0.21 | -0.17 | -0.13 | -0.08 | -0.03 | 0.06 | 0.13 | 0.19 | 0.23 | 0.27 | 0.36 | 0.43 |
| Germany | G0331 | 1 | 0.99 | 0.97 | 0.92 | 0.85 | 0.75 | 0.64 | 0.52 | 0.42 | 0.33 | 0.27 | 0.19 | 0.17 | 0.19 | 0.21 | 0.24 | 0.31 | 0.37 |
| Germany | G0332 | 1 | 0.99 | 0.93 | 0.81 | 0.62 | 0.43 | 0.28 | 0.18 | 0.12 | 0.08 | 0.07 | 0.08 | 0.11 | 0.14 | 0.18 | 0.21 | 0.30 | 0.36 |
| Germany | G0333 | 1 | 0.98 | 0.92 | 0.79 | 0.58 | 0.39 | 0.26 | 0.18 | 0.13 | 0.11 | 0.10 | 0.11 | 0.13 | 0.16 | 0.20 | 0.24 | 0.32 | 0.38 |

**Figure 11. Extract of GloFAS 2.1/2.2 river discharge forecast skill scores for CRPSS against persistence provided for a selection of lead times (LT) out to 30-days ahead as metadata information available through the Climate Data Store (CDS) documentation tab for each of the GloFAS diagnostic points as well as in Supplementary Table 1.**

## 4.4 Future directions

While this paper sets out the components, operational configuration, and a global forecast evaluation of GloFAS 2.1/2.2, the raw real-time forecast and reforecast data have been made openly available to encourage users to use the data for downstream value-added applications and to perform user-specific evaluation of forecast quality. Additionally, GloFAS forecasts and reforecasts have not been post-processed, therefore there is room for users to increase further forecast quality by applying post-processing with their local observations data to correct forecast bias or timing errors, for example. The evaluation carried out here looks at the overall quality of forecasts only. Future work should assess other aspects of forecast quality such as reliability (Robertson et al., 2013), value (Cloke et al., 2017) or performance during extreme events (Bischiniotis et al., 2019). The robust and comprehensive reforecast strategy established for this first evaluation will serve as benchmark against which any new major GloFAS model upgrades can be compared. The GloFAS release strategy now includes public availability and easy access to the river discharge reanalysis, real-time forecasts and reforecasts together with

a first assessment of global forecast skill and will continue for all future major GloFAS launches. We recommend a similar strategy for all global and continental scale hydrological forecasting systems as release of data has traditionally been limited to historical data used for specific inter-comparisons in hydrological performance (e.g. Beck et al., 2017; Towner et al., 2019) rather than a comprehensive set of reforecasts or real-time forecasts. This will pave the way for multi-model forecast skill comparisons, such as those carried out routinely in the NWP field (for example, WMO Lead Centre for Deterministic NWP verification: https://apps.ecmwf.int/wmolcdnv/, last accessed: 13 October 2020).

## 5 Conclusion

It is now technically and computationally feasible to produce operational hydrological forecasting at the global scale. This offers enormous potential in aiding decision-making and humanitarian action in the face of large-scale and often transboundary flood events, as demonstrated by the application of GloFAS in recent floods such as those in Mozambique and Bangladesh. Nevertheless, up until now there have been limited information on hydrological forecast skill, both published in the scientific literature and available to users within the forecast web interface. This paper sets out the model components and operational configuration used in the production of GloFAS real-time forecasts and in the generation of the corresponding large-sample set of 20-year reforecasts. A comprehensive global ensemble forecast evaluation strategy was developed that included a sensitivity assessment on both persistence and climatology benchmark forecasts given the 30-day range of GloFAS (re)forecasts. The analysis shows that on average across the world forecasts should be benchmarked against persistence for lead times up to about two weeks and against climatology for longer lead times. Global forecast skill results shows that GloFAS is skilful in over 93 % of catchments in the short- (1- to 3-days) and medium-range (5- to 15-days) against a persistence benchmark forecast and skilful in over 80 % of catchments out to the extended-range (16- to 30-days) against a climatology benchmark forecast. However, the strength of skill varies considerably by location with GloFAS found to have no or negative skill at longer lead times in broad hydroclimatic regions in tropical Africa, western coast of South America, and catchments dominated by snow and ice in high northern latitudes. These results highlight to users where and when GloFAS is skilful and is a crucial piece of information in the forecast decision-making process and has been made available to forecasters as a new layer in the GloFAS Web Map Viewer for the service-only upgrade to version 2.2 as of 9 December 2020. The results are also useful for model development so that areas where GloFAS performs poorly can be further investigated and new model components designed and tested for improvements. An innovative feature of the GloFAS service development is providing the raw real-time forecast and reforecast data openly to encourage users to explore the data for downstream value-added applications and to perform user-specific and local evaluation of forecast quality.

**Author contributions.** SH and CP designed the study. SH drafted the manuscript and performed the forecast evaluation. EZ developed the GloFAS suites to produce the reforecasts and real-time forecasts. HC and PS helped frame the paper. All co-authors contributed to the editing of the manuscript and to the discussion and interpretation of results.

**Acknowledgments:** Financial support by the Copernicus program, in particular the Copernicus Emergency Management Service (CEMS). We thank Francesca Moschini and Karen O'Regan (both ECMWF) for their help with production of the forecast skill layer and metadata for the GloFAS Web Map Viewer and CDS, respectively; Christopher Barnard (ECMWF) for his help in integrating the real-time forecast and reforecast suits into operations; Corentin Carton De Wiart (ECMWF) for his help in implementing the forecast evaluation suite; and Iacopo Ferrario, Christopher Barnard, Fredrik Wetterhall Sebastien Villaume and the wider CDS/MARS team at ECMWF for their work ingesting the GloFAS data into the CDS and MARS. We thank the water@reading team from the University of Reading who collaborate in the development and use of GloFAS in which many conversations and engagement with users have shaped the service evolution. Finally, we thank Berit Arheimer and two anonymous referees for their constructive feedback that has greatly improved this paper.

**Competing interests.** The authors declare that they have no conflict of interest.

## Appendix A

**Scientific papers and model documentation for the key GloFAS model components. Adapted from Harrigan et al., (2020a)**

| GloFAS component | Description | Reference |
|---|---|---|
| ECMWF IFS | ECMWF Integrated Forecast System (IFS) model documentation for current operational model cycle 47r1 | ECMWF (2020) |
| ERA5 | Global reanalysis dataset using ECMWF IFS model cycle 41r2 from 1979 to present | Hersbach et al. (2020) |
| ECMWF IFS/ERA5 runoff | Surface and sub-surface runoff within ECMWF IFS/ERA5 generated using the HTESSEL land surface model | Balsamo et al. (2009) |
| GloFAS-ERA5 | GloFAS-ERA5 operational global river discharge reanalysis 1979-present | Harrigan et al., (2020a) |
| LISFLOOD river discharge | River discharge generated using LISFLOOD hydrological and channel routing model to route runoff into and through the river network and provide groundwater storage. LISFLOOD includes lake, reservoir and human water use routines | Burek et al. (2013) |
| Lakes and reservoirs used in GloFAS | Incorporated 463 lakes and 667 reservoirs into the GloFAS river network | Zajac et al. (2017) |
| Calibration of LISFLOOD used in GloFAS | LISFLOOD was calibrated against daily river discharge from 1287 observation stations worldwide | Hirpa et al. (2018) |

## Appendix B

```
# Example CDS Python API request script

# Code snippets can be found by clicking 'Show API request' at
# bottom of the download form:
# https://cds.climate.copernicus.eu/cdsapp#!/dataset/cems-glofas-forecast?tab=form

# Instructions on how to download CDS API can be found here:
# https://cds.climate.copernicus.eu/api-how-to

import cdsapi

c = cdsapi.Client()

c.retrieve(
    'cems-glofas-forecast',
    {
        'variable': 'river_discharge_in_the_last_24_hours',
        'format': 'netcdf',
        'product_type': [
            'control_forecast', 'ensemble_perturbed_forecasts',
        ],
        'year': '2020',
        'month': '01',
        'day': '01',
        'leadtime_hour': [
            '24', '48', '72',
            '96', '120', '144',
            '168', '192', '216',
            '240', '264', '288',
            '312', '336', '360',
            '384', '408', '432',
            '456', '480', '504',
            '528', '552', '576',
            '600', '624', '648',
            '672', '696', '720',
        ],
    },
    'download.nc')
```

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
