# Peer review of "Daily ensemble river discharge reforecasts and real-time forecasts from the operational Global Flood Awareness System"

_Hydrology and Earth System Sciences, 2020_

## Referee Comment (RC1) · Anonymous Referee #1 · 22 Nov 2020

Summary

This paper describes the most recent version of the GloFAS ensemble streamflow forecasting system. While there are no major advanced in methods used to generate forecasts, GloFAS is a system of international significance, and highly relevant to readers of HESS. The manuscript is well structured, admirably clear and succinct, and was a pleasure to read. Figures are well presented, and while references are sparse (especially in the introduction), as this paper is essentially focused on presenting an operational system this is ok. As the authors note, a major development is the availability of GloFAS forecast outputs in near-real time, and this is well-explained and documented.

I therefore believe the study ultimately deserves publication. I nonetheless had two major issues with this paper, listed below. I therefore recommend the paper be revised before it can be published.

Major comments

1) There appeared to me to be an error in the calculation of CRPSS with respect to (wrt) persistence - see specific comments below. If this is not due to an error, I would like the authors to explain what to me were counterintuitive results.

2) The authors earmark the assessment of reliability to future work. I do not think this is good enough, given 1) that reliability is a key attribute - in my view at least as important as skill - of ensemble forecasts and 2) their statement in the introduction that "not also having direct access to the raw data precludes the use in further downstream applications (e.g. impact modelling, multi-model forecast systems, production of value-added products for specific sectors such as river transport and hydropower industries, and advancement in techniques requiring large-scale datasets such as machine learning)." This statement implies that the authors expect the outputs in the ways specified - i.e. as direct inputs to impact assessment models of some kind or other. In my experience such models very often require reliable ensembles wrt to observations (or at least unbiased ensembles) as inputs. As GloFAS does not treat hydrological uncertainty, it is highly likely that ensembles are overconfident, particularly at short lead times (e.g. Bennett et al. 2014). I think this is information that users of these outputs, and therefore readers of this paper, would want to know. I therefore would like to see the authors present an assessment of reliability as well as skill, and the ramifications of this assessment discussed. Given the forecasts are likely to be treated as continuous variables in impact models, I suggest using probability integral transforms (PIT, e.g. Gneiting and Katzfuss 2014) to assess reliability (noting the need to generate 'pseudo'-PIT values in cases where streamflow observations can equal zero). If the authors prefer, PIT values can then be summarised with either the alpha-index (Renard et al. 2010) or the beta-score (Keller et al. 2011) (whichever is more suitable) for presentation in plots

similar to Figure 5 or 6.

Specific comments

L88-97 Please provide the model time step at some point in this paragraph.

L125 "https://www.globalfloods.eu/" the hyperlink associated with this text 1) differs from the text and 2) returns a 404 error.

L250 Figure 5. To me, there's something very counterintuitive (and perhaps erroneous?) about the persistence skill plot. The accuracy of persistence (the benchmark, and the demoninator in eq 1) is often very high at short lead times and then declines with lead time - often rapidly. In my experience, this decline is usually much faster than the decline in the accuracy of forecasts. So I would expect CRPSS wrt to persistence to be very low - perhaps even close to 0 - at very short lead times, and then to rise with lead time. But Fig 5 shows the opposite of these trends - i.e. CRPSS wrt persistence starts high and falls with lead time. I can't see how this can occur without a calculation error - though perhaps I've missed something? Even if this is not due to an error, these results at least requires some discussion/explanation. CRPSS calculated wrt to climatology looks sensible to me, which makes the persistence results even more puzzling.

L271 Figure 6 As with Fig 5, I would expect skill wrt to persistence to rise with lead time, not to fall.

L343-345 "Future work should assess other aspects of forecast quality such as reliability (Robertson et al., 2013), value (Cloke et al., 2017) or performance during extreme events (Bischiniotis et al., 2019)." Not suggesting any change here, but the authors may also like to consider calculating the skill/reliability of accumulated volume forecasts (e.g. accumulated 30-day streamflows), as this may well be of interest to reservoir operators and others. The ability to simply sum streamflows of individual ensemble members over various lead times is a major benefit of ensemble streamflow forecasting

systems such as this one (as opposed to probabilistic forecasts generated at discrete lead times).

Typos/grammar/style

L79 "descripted" - 'decribed'?

L140-141 "see for https://confluence.ecmwf.int/display/COPSRV/01.+GloFAS+operational+system a description" should be "see https://confluence.ecmwf.int/display/COPSRV/01.+GloFAS+operational+system for a description

L152-155 "Twice per week ... as real time (Vitart 2014)." Suggest breaking this long sentence in two at the comma.

References

Bennett JC, Robertson DE, Shrestha DL, Wang QJ, Enever D, Hapuarachchi P, Tuteja NK. 2014. A system for continuous hydrological ensemble forecasting (SCHEF) to lead times of 9 days. Journal of Hydrology 519: 2832-2846. DOI: 10.1016/j.jhydrol.2014.08.010.

Gneiting T, Katzfuss M. 2014. Probabilistic forecasting. Annual Review of Statistics and Its Application 1: 125-151. DOI: 10.1146/annurev-statistics-062713-085831.

Renard B, Kavetski D, Kuczera G, Thyer M, Franks SW. 2010. Understanding predictive uncertainty in hydrologic modeling: The challenge of identifying input and structural errors. Water Resources Research 46: W05521. DOI: 10.1029/2009wr008328.

Keller JD, Hense A. 2011. A new non-Gaussian evaluation method for ensemble forecasts based on analysis rank histograms. Meteorologische Zeitschrift 20: 107-117. DOI: 10.1127/0941-2948/2011/0217.

---

## Referee Comment (RC2) · Berit Arheimer (Referee) · 17 Dec 2020

**Referee comments on "Daily ensemble river discharge reforecasts and real-time forecasts from the operational Global Flood Awareness System" by Harrigan et al., 2020 in HESSD.**

This paper shows recent progress in global river forecasts from the Glofas modelling system. Such data is indeed very useful and appreciated by many users at the global scale, especially by low- and middle-income countries who might not have access to their own river-forecast system. Accordingly, it is very important to evaluate such systems scientifically before launching them operationally.

The paper gives a very good overview of a river-discharge forecast system, which is indeed valuable for the scientific community to learn more about, as such systems are dedicated to national/international institutes with advanced IT infrastructure and operational production.

My main concern with this paper is that I miss a scientific question and the story of what kind of new scientific knowledge we have learnt from using the forecasting system and evaluation method described.

The Glofas model and forecasting system has been described before in the scientific literature and the focus of this paper seem to be that the results are now part of the climate service C3S, but this is hardly a scientific finding. New datasets should rather be published in ESSD, in which Glofas results have already been published. Likewise, the methods used for forecast evaluation are standard and has been published before. For publications in HESS I expect a more scientific analysis of the results and conclusions about new knowledge from the identified scientific achievements with impact on our understanding of Hydrology or Earth Systems. Right now, I have difficulties to find a clear take-home message in the current version of this paper. It is very descriptive and less analytic.

I therefore suggest to find a scientific angle from current discussions in the research community and tell the story of the results from that perspective.

Interesting scientific questions could for instance be:

- On the method side: How should we evaluate forecasts – what metrics are there, how do they compare and what does different metrics contribute in understanding/reliability for the user community and research community, respectively?
- Could the metrics presented (and argued for?) in this paper be compared with other metrics, to show their excellence and benefits to users/scientists? (is there a take-home message or guide-lines to the scientific community from using a specific metric/evaluation method compared to another?) What are the options?
- On the understanding of hydrology: what are the attributes for catchments/regions with high or low skills in forecasting? i.e. which processes do we need to learn more about to improve the quality of river-discharge forecasts?
- How does different global river-discharge forecast systems compare to each other? Can we learn from different model setups and elaborations on procedures, process descriptions or geophysical representation?

Please, find some detailed comments on current manuscript below. Apologies for mentioning my own work, but I am very eager to start comparing model results at the global scale soon. 😊

**Introduction**

Line 31: Reference Blöschl, et al. 2019 does not evaluate risks or hazards.

Line 37: also note the global and continental scale forecasting based on sharing the world-wide HYPE model:

> *Arheimer, B., Pimentel, R., Isberg, K., Crochemore, L., Andersson, J. C. M., Hasan, A., and Pineda, L., 2020. Global catchment modelling using World-Wide HYPE (WWH), open data and stepwise parameter estimation, Hydrol. Earth Syst. Sci. 24, 535–559, https://doi.org/10.5194/hess-24-535-2020*

Line 60-70: In fact, global river forecasts and reforecasts are also available at https://hypeweb.smhi.se/ where the user can subscribe to seasonal forecasts with monthly data. In addition, 1-10 days forecasts with thresholds based on return periods of high flows can be ordered at https://hypeweb.smhi.se/water-services/data-delivery-services/

Section 2. even though Glofas has been evaluated against observed river flow in previous publications, it would be helpful to include such information about model performance vs absolute values also. For instance, Fig 4 could also include colors of KGE performance (modelled values vs observed values) in the circles showing location of gauges. This would make this figure much more informative and help the reader a lot to judge model performance. Please, check the color coding in Arheimer et al., so the overall pattern of model performance could be compared. Please, also mention median KGE at global scale (no you only say that it was skillful, which is very vague).

Section 3: please start with some sentences summarizing the evaluation concept – e.g. that you use scores with met. model vs observed met. model ("a perfect weather model") and correlation with observations. It would also be interesting for many users to actually see some scores to absolute values as well – or at least to discuss the difficulties here.

Section 4: the Glofas results could be compared with results from another model, using the same metrics across Europe, presented by:

> *Pechlivanidis, I. G., Crochemore, L., Rosberg, J., & Bosshard, T. (2020). What are the key drivers controlling the quality of seasonal streamflow forecasts? Water Resources Research, 56, e2019WR026987. https://doi.org/10.1029/2019WR026987*

To further explore and evaluate the added value of the Glofas system, it could also be compared to warning issued by National forecast services for specific regions or countries, or to soft information from new items reporting floods, to check if the alerts actually captured something real.

Line 265: Attribution is also discussed in the above-mentioned paper. It is another interesting scientific analysis, which deserves much more attention – also in this global study of model performance. Such an analysis would make this paper much more scientifically interesting.

I am looking forward to read a new more elaborated version of this paper, with a scientific discussion linked to the methodological description.

---

## Referee Comment (RC3) · Anonymous Referee #3 · 23 Dec 2020

This paper described the development, application and user service upgrade of GLO-FAS to include reforecasts and reforecast-based skill calculations. It probably contains more engineering-related detail than is typical in a HESS paper, but it does advance the science as well in providing a working example of the applied science concept that reforecasts can help to usefully complement the information available from forecasts alone – thus I think it is appropriate for HESS. It would be nice to see a bit more framing on where a forecast effort such as GLOFAS fits in overall field of hydrologic forecasting, but the authors may deem that after some years of running GLOFAS, it is now a widely understood / accepted approach, and that the overall GLOFAS rationale is adequately addressed in prior papers. I don't actually think it is as well understood outside

of Europe, where EFAS served as an introduction to this type of service. Partly for that reason, I'd suggest that the authors do more to highlight some expected limitations of GLOFAS relative to a local/regional forecast (suggestions below), particularly given the use of perfect model benchmarks. On the plus side, the paper could strengthen the context framing by noting that it represents one of the first large operational scale effort at reforecasting in hydrology, in contrast to the introduction of reforecasting more broadly for weather and climate over 10 years ago. Overall, however, I find the paper to be a high-quality, very readable effort, presenting a range of accessible and useful information, hence I recommend publication with relatively minor clarifications and adjustments listed below.

Specific comments:

53: 'originally designed for large/transboundary river basins' – this is surprising because those would almost all be regulated and impaired, yet glofas does not represent such effects. could this statement be sharpened? if glofas can't be expected to forecast mainstem flows in such basins accurately, what was glofas designed to do more specifically? eg forecast runoff anomalies in such basins? or smaller tributaries across large basin domains? or natural flow changes and risks?

58: there are more service-related gaps, I think. at the end of this paragraph could another sentence be added to suggest that 'Other concerns include ...' where I could imagine that the lack of information about where glofas reflects upstream management effects might be another one. ideally a user could mask out reaches where it's thought that what is shown cannot represent the reality of the river flow, perhaps because it is 50% determined by a reservoir release or major diversion.

Fig1: The label 'hydrological model' is interesting because most of the hydrological components of LISFLOOD are not used – really it is the empirical groundwater attenuation and the channel routing of runoff. A more descriptive label for this component might be 'Catchment and channel routing' (whether GW or channel it's all a kind of

routing, conceptually). Not a big issue but it may be confusing given that a lot of hydrology (runoff generation, snow accum/melt, et) is done by the LSM. btw, LISFLOOD could conceivably also offer a hillslope routing function (gamma or UH distribution).

114: Would be helpful to add a sentence to clarify how these reservoirs are represented and their realism – eg level-pool scheme, fixed rule curve

174: Perhaps add a sentence or two here or in the later discussion if the reanalysis latency impacts the skill of real-time forecasts (2-5 days is a lot of lag for a flood forecast!) or means that the skill of the hindcasts may be systematically higher than that of the real-time forecasts, since presumably the reanalysis initialization in the past doesn't have latency issues.

178: Can you say what determines 'skillful'? eg 5% kge above benchmark in the SS?

199, 235: Perhaps add a sentence explaining what fraction of these sites are impaired/unimpaired and what 'synthetic' means in the context of an observation (eg labeled in the table). In general, it should be quite clear in the paper when you are verifying against the real versus perfect model world. The initial description of verification against in situ gage stations may slightly obscure the later default to benchmarking against the reanalysis discharge. Could a sentence be added at 235 to state what is gained/lost in the interpretation of skill through comparing to the perfect model benchmark?

239: again could you quantify in a sentence what you consider 'skillful'. without going fully statistical (ie significance level of a skill scores X% above 0), what is the rule of thumb used by the authors to consider a forecast 'skillful'?

Fig 5: It's curious that the persistence-benchmarked forecast SS is so high at timestep 1. If anything I would expect it to start slightly lower than its peak SS because as you move from longer leads to approach T0 the persistence forecast, in theory at least, approaches zero error, and the persistence and actual forecasts approach each other.

There must be something in practice here that means this is not the case, ie persistence has an offset at T0 that is not present in the actual forecast. Especially in medium to large rivers, much of the time by day 1 the flow from T0 has not changed much. Is it correct that the actual forecast is 95% better than persistence even at day1? Can/should this behavior be explained in the paper?

343: I don't quite agree, as I think of bias & mean error-based scores reflecting only accuracy. The use of an integrated score such as crps means the results are sensitive not just to accuracy but to forecast spread (hence reliability). Perhaps rephrase here?

354: It would be helpful to add a discussion returning to some of the major caveats on the applicability of the analysis. I strongly support the overall message of the paper and commend the effort for having generated reanalyses and used them to demonstrate the kind of information one can derive from them, but I also recognize that in many areas the skill / usability estimates are compromised by use of perfect model benchmarks, and the lack of representation of real-world impairments in GLOFAS. Could the authors walk us through some of the possible limitations, eg in a paragraph, discuss a few cases that users may face in interpreting this kind of skill or using the reforecasts? eg, in the best case, through the model may have some biases it generally represents catchment variability such that the perfect model skill analysis is more or less directly transferrable to real world conditions. A use could infer usability qualitatively or even apply quantitative post-processing methods for their own site observations to adjust the skill scores accordingly. In a medium case, the model is badly biased (say magnitudes by 50-100% and seasonal timing off by 30 days), but still represents observed variability in high/low flow conditions – what more would be required of a user then? And in the worst case, say on highly regulated rivers, perhaps glofas cannot be used except in the most extreme situations, eg when there is so much or so little water that management effects are secondary. Would this suggest any other future directions, eg providing guidance or tools on such user-based post-processing and analysis?

---

## Author Comment (AC3) · 20 Jan 2021

Please find our initial response to RC3 in the attached. Kind regards, Shaun Harrigan on behalf of all co-authors

Please also note the supplement to this comment: https://hess.copernicus.org/preprints/hess-2020-532/hess-2020-532-AC3-supplement.pdf

---

## Author Response (AR1)

Dear Jim,

Please find below our responses to the three reviewers. We highlight where we have made changes to the resubmitted manuscript considering their comments. As a general point, the number of citations, particularly in the introduction and discussion sections, were too few in the original submission (as highlighted by you when we first submitted our paper). We have therefore increased the breadth of research cited.

Kind regards,
Shaun Harrigan on behalf of all co-authors

**Response to RC1**

*Anonymous Referee #1 (R#1)'s original text in* black *with our initial response in* blue*.*

Summary
This paper describes the most recent version of the GloFAS ensemble streamflow forecasting system. While there are no major advanced in methods used to generate forecasts, GloFAS is a system of international significance, and highly relevant to readers of HESS. The manuscript is well structured, admirably clear and succinct, and was a pleasure to read. Figures are well presented, and while references are sparse (especially in the introduction), as this paper is essentially focused on presenting an operational system this is ok. As the authors note, a major development is the availability of GloFAS forecast outputs in near-real time, and this is well-explained and documented.

I therefore believe the study ultimately deserves publication. I nonetheless had two major issues with this paper, listed below. I therefore recommend the paper be revised before it can be published.

We thank the reviewer for their positive words about our manuscript and constructive comments below.

Major comments
1) There appeared to me to be an error in the calculation of CRPSS with respect to (wrt) persistence - see specific comments below. If this is not due to an error, I would like the authors to explain what to me were counterintuitive results.

We provide a detailed explanation of the pattern of CRPSS wrt persistence below at your specific comment and show the results are as expected.

2) The authors earmark the assessment of reliability to future work. I do not think this is good enough, given 1) that reliability is a key attribute - in my view at least as important as skill - of ensemble forecasts and 2) their statement in the introduction that "not also having direct access to the raw data precludes the use in further downstream applications (e.g. impact modelling, multi-model forecast systems, production of value-added products for specific sectors such as river transport and hydropower industries, and advancement in techniques requiring large-scale datasets such as machine learning)." This statement implies that the authors expect the outputs in the ways specified - i.e. as direct inputs to impact assessment models of some kind or other. In my experience such models very often require reliable ensembles wrt to observations (or at least unbiased ensembles) as inputs. As GloFAS does not treat hydrological uncertainty, it is highly likely that ensembles are overconfident, particularly at short lead times (e.g. Bennett et al. 2014). I think this is information that users of these outputs, and therefore readers of this paper, would want to know. I therefore would like to see the authors present an assessment of reliability as well as skill, and the ramifications of this assessment discussed. Given the forecasts are likely to be treated as continuous variables in impact models, I suggest using probability integral transforms (PIT, e.g. Gneiting and Katzfuss 2014) to assess reliability (noting the need to generate 'pseudo'-PIT values in cases where streamflow observations can equal zero). If the

authors prefer, PIT values can then be summarised with either the alpha-index (Renard et al. 2010) or the beta-score (Keller et al. 2011) (whichever is more suitable) for presentation in plots similar to Figure 5 or 6.

We agree that reliability is indeed an important aspect of hydrological forecast quality, but there are many important aspects of forecast quality relevant for GloFAS users (for example, forecast skill for extreme events). The focus of this paper (as outlined in L70-72) is to provide a detailed description of how the GloFAS forecast datasets (both real-time and reforecasts) are generated and present a first global assessment of ensemble forecast skill against two of the most common benchmarks in the hydrological literature; thus defining the new GloFAS "headline skill score" that is added as a new layer on the GloFAS web interface. We do not claim that this first evaluation covers all forecast quality aspects, nor do we believe this specific paper is the right place to add the evaluation of just reliability and not other aspects that might be important for the diverse range of users. By providing the large-sample reforecasts dataset openly, we strongly encourage users of GloFAS foreacsts to conduct their own specific and local evaluation. We see this paper as the first step, and as part of the ongoing GloFAS evolution will expand global-scale evaluation efforts to other forecast quality aspects. Thank you for your suggestion on the method for evaluation of reliability, we will certainly take this into consideration going forward.

Specific comments
L88-97 Please provide the model time step at some point in this paragraph.

The ECMWF ENS is run at a 6-hourly forecast time step and for ingestion into the GloFAS hydrological modelling chain, data from the 00 UTC run is extracted and aggregated to 24-houly time step. This sentence has been added to the end of this paragraph in Sect. 2.1.

L125 "https://www.globalfloods.eu/" the hyperlink associated with this text 1) differs from the text and 2) returns a 404 error.

Thank you for noticing this. The hyperlink should have pointed to the main GloFAS web site: "https://www.globalfloods.eu/" and has been corrected in the resubmitted manuscript.

L250 Figure 5. To me, there's something very counterintuitive (and perhaps erroneous?) about the persistence skill plot. The accuracy of persistence (the benchmark, and the demoninator in eq 1) is often very high at short lead times and then declines with lead time - often rapidly. In my experience, this decline is usually much faster than the decline in the accuracy of forecasts. So I would expect CRPSS wrt to persistence to be very low perhaps even close to 0 - at very short lead times, and then to rise with lead time. But Fig 5 shows the opposite of these trends - i.e. CRPSS wrt persistence starts high and falls with lead time. I can't see how this can occur without a calculation error - though perhaps I've missed something? Even if this is not due to an error, these results at least requires some discussion/explanation. CRPSS calculated wrt to climatology looks sensible to me, which makes the persistence results even more puzzling.

We do not think the results are counterintuitive, but agree it is very useful to have access to the individual components of the skill score equation 1 (i.e. $CRPS_{fc}$ and $CRPS_{bench}$) to better interpret the accuracy of the GloFAS forecasts *relative* to the accuracy of both persistence and climatology benchmark forecasts, as a function of lead time. We therefore show in Fig. A (below) the individual CRPS accuracy components in equation 1 (i.e. CRPS of GloFAS forecasts (black line), persistence benchmark (blue line) and climatology (red line)) as a median across n=5997 river points. This plot helps interpret the global median CRPSS results (solid lines) presented in Figure 5 in the original manuscript. While the CRPSS in Fig. 5 is dimensionless with an optimum value of 1, the CRPS error is measured in units of the variable being evaluated (here $m^3/s$) and so has an optimum of 0 (i.e. perfect accuracy against the reanalysis).

It is clear from Fig. A and consistent with your comment: the accuracy of persistence (blue line) is highest at short lead times then gets rapidly less accurate as lead time increases. But it is also the case that the accuracy of GloFAS forecasts (black line) are highest at short lead times, and get less accurate as lead time increases. The decline of accuracy of persistence is however faster than the decline of the accuracy of GloFAS forecasts; this is also consistent with your comment. The persistence benchmark is very simple (use the reanalysis river discharge value from day before the forecast for all lead times), whereas the GloFAS forecast includes both information on the initial conditions as well as meteorological forecast information. Therefore, the accuracy of the GloFAS forecasts is (on average) higher than persistence, even for short lead times. In our opinion this is intuitive, and it would be more surprising if the considerably more sophisticated GloFAS forecasts were only as accurate or marginally more accurate than a simple persistence forecast, including at short lead times. Further, the core meteorological variables that drive GloFAS forecasts (e.g. precipitation and temperature) are most accurate at short lead times (Haiden et al., 2019). We also include the CRPS for climatology (red line) in Fig. A for completeness.

To help users better interpret the CRPSS for their stations of interest, we have included CRPSS as well as CRPS plots as a clickable "pop out" window as part of the new GloFAS "forecast skill" layer on the GloFAS Web Map Viewer for the release of version 2.2 on 2 December 2020 (https://confluence.ecmwf.int/display/COPSRV/Latest+operational+release%3A+GloFAS+v2.2).

We have added Figure A as an additional panel - now Figure 5b in the resubmitted manuscript. To reflect the update to the CRPSS/CRPS plots within the "Forecast skill" layer on the GloFAS website, we have updated Sect. 4.2 and Fig. 9 in the resubmission.

[Figure]

**Figure A (new Figure 5b).: Global median Continuous Ranked Probability Score (CRPS) for GloFAS forecasts (black line) and both persistence (blue line) and climatology (red line) benchmark forecasts from 1- to 30-day lead times with respect to GloFAS-ERA5 river discharge reanalysis across 5997 diagnostic river points.**

L271 Figure 6 As with Fig 5, I would expect skill wrt to persistence to rise with lead time, not to fall.

As per the response to comment above, the accuracy of both the persistence benchmark forecast and GloFAS forecasts themselves decrease with lead time, but the accuracy of GloFAS forecasts decrease at a slower rate (Figure A above).

L343-345 "Future work should assess other aspects of forecast quality such as reliability (Robertson et al., 2013), value (Cloke et al., 2017) or performance during extreme events (Bischiniotis et al., 2019)." Not suggesting any change here, but the authors may also like to consider calculating the skill/reliability of accumulated volume forecasts (e.g. accumulated 30-day streamflows), as this may well be of interest to reservoir operators and others. The ability to simply sum streamflows of individual ensemble members over various lead times is a major benefit of ensemble streamflow forecasting systems such as this one (as opposed to probabilistic forecasts generated at discrete lead times).

We currently we use weekly river discharge averages within GloFAS-Seasonal operationally and for seasonal forecast evaluation (see Emerton et al., 2018). However, providing forecast products (and forecast evaluation) at a range of different accumulations is something we will take on board and seek feedback from GloFAS users, thank you for your comment.

Typos/grammar/style

L79 "descripted" - 'decribed'?

This should have been "described" and has been updated in the resubmitted manuscript.

L140-141 "see for https://confluence.ecmwf.int/display/COPSRV/01.+GloFAS+operational+system a description" should be "see https://confluence.ecmwf.int/display/COPSRV/01.+GloFAS+operational+system for a description

This has been updated in the resubmitted manuscript.

L152-155 "Twice per week ... as real time (Vitart 2014)." Suggest breaking this long sentence in two at the comma.

We have broken this long sentence into two in the resubmission:

"A reforecast task is run twice per week (on Mondays and Thursdays) in parallel to the real-time forecast, using ERA5 atmospheric reanalysis (Hersbach et al., 2020) for initial conditions of past dates. A reforecast of the corresponding date for the previous 20 years is produced with a reduced number of 11 ensemble members but using the same model version as real-time (Vitart, 2014)."

Thank you again for your insight and we appreciate your time to review our manuscript,

Kind regards,
Shaun Harrigan on behalf of all co-authors

**Response to RC2**

*Berit Arheimer (R#2)'s original text in* black *with our initial response in* blue*.*

This paper shows recent progress in global river forecasts from the Glofas modelling system. Such data is indeed very useful and appreciated by many users at the global scale, especially by low- and middle-income countries who might not have access to their own river-forecast system. Accordingly, it is very important to evaluate such systems scientifically before launching them operationally.

The paper gives a very good overview of a river-discharge forecast system, which is indeed valuable for the scientific community to learn more about, as such systems are dedicated to national/international institutes with advanced IT infrastructure and operational production.

We thank Berit for her positive words about our manuscript and constructive feedback and suggestions that have helped refine our paper.

My main concern with this paper is that I miss a scientific question and the story of what kind of new scientific knowledge we have learnt from using the forecasting system and evaluation method described.

The Glofas model and forecasting system has been described before in the scientific literature and the focus of this paper seem to be that the results are now part of the climate service C3S, but this is hardly a scientific finding. New datasets should rather be published in ESSD, in which Glofas results have already been published. Likewise, the methods used for forecast evaluation are standard and has been published before. For publications in HESS I expect a more scientific analysis of the results and conclusions about new knowledge from the identified scientific achievements with impact on our understanding of Hydrology or Earth Systems. Right now, I have difficulties to find a clear take-home message in the current version of this paper. It is very descriptive and less analytic.

We take on board your point that the take-home messages in the current version could be clearer and will ensure they are sharper in the resubmitted manuscript. The original pre-operational GloFAS (version 1) was indeed described by Alfieri et al. (2013), however we disagree with your point that the scientific details of the fully operational GloFAS (version 2) have already been published in the hydrological scientific literature. Our manuscript is the first time the fully operational real-time forecast configuration has been published. Uniquely, this is the first time the large-sample and long-term reforecast strategy we use for generating and evaluating the forecast skill of GloFAS has ever been published, and we think this provides an important advancement in the area of global hydrological forecasting as well as users of GloFAS forecasts. We do not claim the evaluation method is new, but what is novel is the scale of the evaluation (both in space and across the long forecast range) and how the data and results are delivered to the hydrological community. Additionally, to our knowledge (apologies if incorrect), no other global operational hydrological forecasting system currently provides such a long-term and large-sample set of reforecasts, delivered in a free and open way, with summary evaluation results available on the user interface – we think this is a significant advancement towards transparency and as such, this procedure will be implemented with each new major release of GloFAS; we very much encourage other systems to follow in this direction. Once large sets of ensemble reforecasts are available from multiple forecast systems, even more interesting scientific questions around predictability can be uncovered, and we look forward to future collaborations with your group and WWH in this regard!

Our paper is not simply a data paper so we disagree ESSD is the most appropriate outlet. We want to communicate with the wider hydrological science community our scientific description of the GloFAS configuration, our method for evaluating the skill of GloFAS, and the findings of when and where in the world GloFAS ensemble forecasts are skilful against the two primary benchmark forecasts used in hydrological forecasting. HESS in particular has become a key journal for publishing papers in the area of hydrological forecasting and therefore we hope the editor agrees is a good home.

I therefore suggest to find a scientific angle from current discussions in the research community and tell the story of the results from that perspective.

Interesting scientific questions could for instance be:

- On the method side: How should we evaluate forecasts – _what metrics are there, how do they compare and what does different metrics contribute in understanding/reliability for the user community and research community, respectively?

- Could the metrics presented (and argued for?) in this paper be compared with other metrics, to show their excellence and benefits to users/scientists? (is there a take-home message or guide-lines to the scientific community from using a specific metric/evaluation method compared to another?) What are the options?

- On the understanding of hydrology: what are the attributes for catchments/regions with high or low skills in forecasting? i.e. which processes do we need to learn more about to improve the quality of river-discharge forecasts?

- How does different global river-discharge forecast systems compare to each other? Can we learn from different model setups and elaborations on procedures, process descriptions or geophysical representation?

Thank you for this valuable list of avenues for further research. Most of these would be very interesting standalone studies in their own right. We agree with your bullet point 3 that the results of forecast skill could be expanded in the revised manuscript to provide more insight into the regions/catchments with with high or low skill.

To address this we have added a new Figure 8 (below) to the revised manuscript. The new figure shows GloFAS forecast skill (CRPSS) by latitude (a), catchment area (b) and the Richards-Baker Flashiness Index (c). Results have been worked into the existing results text in Sect. 4.2.2 and a new section 4.2.3 titled 'GloFAS skill by catchment area and hydrological flashiness' has been added. These new results together with a more broad discussion of the literature have led us to focus future research and development of GloFAS in three main areas to improve the quality of river discharge forecasts: i.) the need for higher spatial and temporal resolution hydrological modelling, ii.) the need to improve precipitation forecasts within the ECMWF global weather model, especially for convective events in the tropics and how that might improve hydrological forecasting, and iii.) further investigation into the representation of snow processes and their impact on forecast skill.

[Figure]

**New Figure 8: GloFAS 2.1/2.2 Continuous Ranked Probability Skill Score (CRPSS) for 5-day (green dots; against persistence benchmark) and 20-day (brown dots; against climatology benchmark) lead times at 5997 diagnostic river points by degree latitude of the river point (a), catchment area (b), and RB Flashiness index (c). Spearman Rank correlation coefficients (Rho) for each combination given in text in the bottom right.**

Please, find some detailed comments on current manuscript below. Apologies for mentioning my own work, but I am very eager to start comparing model results at the global scale soon. 😊

Yes indeed, we are already collaborating on a comparison between GloFAS and World-Wide HYPE in terms of hydrological simulation performance, but the work is not yet completed. We also agree that the prospect of being able to compare multiple operational global forecasting systems (and not only GloFAS and WWH, but others as well) in terms of ensemble forecast skill would provide extremely valuable information to the scientific and forecast user community and are too are very eager to participate further.

**Introduction**

Line 31: Reference Blöschl, et al. 2019 does not evaluate risks or hazards.

This sentence has been modified to "Hydrological extremes, such as floods and droughts, have severe negative socio-economic impacts and climate change is expected to alter their timing and magnitude (Blöschl et al., 2017, 2019; Ward et al., 2020)"

Line 37: also note the global and continental scale forecasting based on sharing the world-wide HYPE model:

> *Arheimer, B., Pimentel, R., Isberg, K., Crochemore, L., Andersson, J. C. M., Hasan, A., and Pineda, L., 2020. Global catchment modelling using World-Wide HYPE (WWH), open data and stepwise parameter estimation, Hydrol. Earth Syst. Sci. 24, 535–559, https://doi.org/10.5194/hess-24- 535-2020*

Thank you for the suggested paper. We have added it to this section in the resubmitted manuscript.

Line 60-70: In fact, global river forecasts and reforecasts are also available at https://hypeweb.smhi.se/ where the user can subscribe to seasonal forecasts with monthly data. In addition, 1-10 days forecasts with thresholds based on return periods of high flows can be ordered at https://hypeweb.smhi.se/water-services/data-delivery-services/

Thank you for this information. GloFAS has been providing an on-demand tailored user data service free of charge since it went pre-operational in 2011, but the present manuscript outlines a step-change in the service of large scale GloFAS data. What would be fantastic for the community is to be able to access sets of reforecasts generated from many hydrological forecast centres in a standardised format and central data service portal. This has been common practice in the weather and climate fields for years and has really facilitated advancement in forecasting science.

Section 2. even though Glofas has been evaluated against observed river flow in previous publications, it would be helpful to include such information about model performance vs absolute values also. For instance, Fig 4 could also include colors of KGE performance (modelled values vs observed values) in the circles showing location of gauges. This would make this figure much more informative and help the reader a lot to judge model performance. Please, check the color coding in Arheimer et al., so the overall pattern of model performance could be compared. Please, also mention median KGE at global scale (no you only say that it was skillful, which is very vague).

Given the full evaluation of the hydrological model performance of GloFAS is published in another paper (Harrigan et al., 2020) we only summarised the results briefly here and pointed the reader to the original paper where the detailed statistics (such as median KGE at the global scale, which is 0.31) and indeed all raw statistics for each of the 1801 stations were provided within the Supplementary Information of Harrigan et al. (2020) to allow for comparison.

However, we agree that having the hydrological model performance results summarised in Figure 4 would help the reader would be help for the reader in the present manuscript. We believe it is most helpful to do so using the modified KGE as a skill score (KGESS) against a mean flow benchmark, following Knoben et al. (2019) as done in Harrigan et al. (2020), and have updated Figure 4 in the resubmitted manuscript (also shown below).

[Figure]

**Updated Figure 4. GloFAS diagnostic river points (n=5997) are highlighted by grey dots. Coloured dots show hydrological performance of GloFAS-ERA5 river discharge reanalysis against a subset of GloFAS diagnostic river points with observations (n=1801) from Harrigan et al. (2020) using the modified Kling–Gupta efficiency skill score (KGESS). Optimum value of KGESS is 1. Blue (red) dots show catchments with positive (negative) hydrological skill.**

Section 3: please start with some sentences summarizing the evaluation concept – _e.g. that you use scores with met. model vs observed met. model (_ "a perfect weather model") and correlation with observations. It would also be interesting for many users to actually see some scores to absolute values as well – or at least to discuss the difficulties here.

We mention explicitly that the forecast skill using the CRPSS (i.e. Sect. 3.3) is "verified against GloFAS-ERA5 river discharge reanalysis used as proxy observations [or 'perfect model'] (following Alfieri et al., 2014)" in L254-235. We have made it clear that both the benchmark forecasts and the verifying observations are based on the river discharge reanalysis rather than in situ station observations. This approach is common practice in forecast evaluation (e.g. as in Pechlivanidis et al. (2020) mentioned below), but for the benefit of a broader audience we have added the justification of this approach at the end of Sect. 3.3 in the resubmitted manuscript as follows:

"Calculating forecast skill against proxy observations such as reanalysis is common in hydrological forecasting as it has the advantage of providing a spatiotemporally complete picture of forecast skill, currently not possible based on availability of the current global in situ observed river network (Lavers et al., 2019). It also allows the forecast predictability range to be isolated in the absence of systematic hydrological model errors. There is a disadvantage of forecast evaluation against proxy observations for catchments that represent hydrological dynamics poorly. While Harrigan et al. (2020a) demonstrate the performance of GloFAS-ERA5 reanalysis is largely hydrologically skilful, readers should be aware that there are areas where performance is poor and that there are large parts of the world where the performance is unknown due to the lack of in situ observations to evaluate against (Figure 4)."

Section 4: the Glofas results could be compared with results from another model, using the same metrics across Europe, presented by:

*Pechlivanidis, I. G., Crochemore, L., Rosberg, J., & Bosshard, T. (2020). What are the key drivers controlling the quality of seasonal streamflow forecasts? Water Resources Research, 56, e2019WR026987. https://doi.org/10.1029/2019WR026987*

One of the key benefits of providing the GloFAS data openly and free of charge on the Copernicus Climate Data Store (CDS) is that it now facilitates further scientific evaluation and inter-comparisons of similar forecasting systems offering their data in the same way. We however think a more appropriate comparison of GloFAS forecasts with Pechlivanidis et al. (2020) is to undertake it with the GloFAS-Seasonal system (Emerton e a., 2018), which is forced by the SEAS5 climate output. All GloFAS-Seasonal data including a comprehensive set of reforecasts are now available through the CDS: https://cds.climate.copernicus.eu/cdsapp#!/dataset/cems-glofas-seasonal-reforecast?tab=overview; for a higher resolution hydrological seasonal forecasting system at European scale, the EFAS-Seasonal complete dataset is also available from the CDS (https://cds.climate.copernicus.eu/cdsapp#!/dataset/efas-seasonal-reforecast?tab=overview).

To further explore and evaluate the added value of the Glofas system, it could also be compared to warning issued by National forecast services for specific regions or countries, or to soft information from new items reporting floods, to check if the alerts actually captured something real.

Evaluation of flood events against a wider set of observations is a very good idea and something that will be expanded in future assessments, but is outside the scope of this first evaluation to determine overall ensemble forecast skill against the two key scientific benchmark forecasts: persistence and climatology.

Line 265: Attribution is also discussed in the above-mentioned paper. It is another interesting scientific analysis, which deserves much more attention – _also in this global study of model performance. Such an analysis would make this paper much more scientifically interesting.

This point is related to the one you bring up in the above bullet list of interesting further scientific questions. We agree and have now expanded section 4.2 and include new Figure 8 (see response above) in the resubmitted version.

I am looking forward to read a new more elaborated version of this paper, with a scientific discussion linked to the methodological description.

We appreciate your time to review our manuscript and thank you again for your constructive feedback and many ideas for further research and collaborations!

Kind regards,
Shaun Harrigan on behalf of all co-authors

**Response to RC3**

*Anonymous Referee #3 (R#3)'s original text in* black *with our initial response in* blue.

This paper described the development, application and user service upgrade of GLOFAS to include reforecasts and reforecast-based skill calculations. It probably contains more engineering-related detail than is typical in a HESS paper, but it does advance the science as well in providing a working example of the applied science concept that reforecasts can help to usefully complement the information available from forecasts alone – thus I think it is appropriate for HESS. It would be nice to see a bit more framing on where a forecast effort

such as GLOFAS fits in overall field of hydrologic forecasting, but the authors may deem that after some years of running GLOFAS, it is now a widely understood / accepted approach, and that the overall GLOFAS rationale is adequately addressed in prior papers. I don't actually think it is as well understood outside of Europe, where EFAS served as an introduction to this type of service. Partly for that reason, I'd suggest that the authors do more to highlight some expected limitations of GLOFAS relative to a local/regional forecast (suggestions below), particularly given the use of perfect model benchmarks. On the plus side, the paper could strengthen the context framing by noting that it represents one of the first large operational scale effort at reforecasting in hydrology, in contrast to the introduction of reforecasting more broadly for weather and climate over 10 years ago. Overall, however, I find the paper to be a high-quality, very readable effort, presenting a range of accessible and useful information, hence I recommend publication with relatively minor clarifications and adjustments listed below.

We thank the reviewer for their positive words about our manuscript and constructive comments.

We agree very much with you on the need to frame the justification/reason for a system such as GloFAS in the context of local systems and within the overall field of hydrological forecasting - thanks for the suggestion! GloFAS aims to provide complementary information in addition to, rather to instead of, locally calibrated catchment forecast systems. In the first instance a global system is useful for providing a hydrological forecast in regions where there is currently no operational local system, or when a local system covers only part of a larger, often transboundary basin. Further, there are users that require a global overview of potential upcoming extreme events, such as international disaster and humanitarian agencies.

We have expanded the introduction section to include the following:

"GloFAS is not designed to be a replacement for local operational hydrological forecasting systems; in many parts of the world however a local or national system for operational forecasts of river discharge does not yet exist so it might be the only information available. GloFAS covers all river basins out to medium- and extended-range lead times (30-days ahead) and updated daily, with GloFAS-Seasonal (Emerton et al., 2018) updated monthly out to a 16-week lead time. Therefore, it has been used to complement local forecast systems by allowing forecasters to gain information on surrounding and upstream basins, monitoring for potential flood signals where advanced warning is needed."

Specific comments:
53: 'originally designed for large/transboundary river basins' – this is surprising because those would almost all be regulated and impaired, yet glofas does not represent such effects. could this statement be sharpened? if glofas can't be expected to forecast mainstem flows in such basins accurately, what was glofas designed to do more specifically? eg forecast runoff anomalies in such basins? or smaller tributaries across large basin domains? or natural flow changes and risks?

As mentioned in L114-115, 667 of the largest reservoirs are represented in GloFAS. Nevertheless, the reservoir scheme is of course a simplification of reality and the actual real-time release schedules of individual reservoirs is sensitive information and typically not publicly available. However, to help guide forecasters, the locations of the reservoirs explicitly modelled in GloFAS together with the ratio of reservoir volume to mean annual discharge for all downstream river cells are provided to users as supporting map layers within the GloFAS Web Map Viewer (https://www.globalfloods.eu/). While GloFAS generates raw river discharge magnitudes, the nature of a global-scale system means it must rely openly available datasets and is run typically at coarser resolution than locally calibrated models, thus provides varying degrees of accuracy with significant biases, as documented in Harrigan et al. (2020). Nevertheless, GloFAS forecasts are compared relative to thresholds derived from the same model, and therefore can provide awareness of anomalously high river discharge. For example, if 80 % of forecasted ensemble members exceeded the 1 in 20-year modelled threshold then this would signify an extreme forecast 'signal' irrespective to any systematic biases in the hydrological model.

58: there are more service-related gaps, I think. at the end of this paragraph could another sentence be added to suggest that 'Other concerns include : : :' where I could imagine that the lack of information about where glofas reflects upstream management effects might be another one. ideally a user could mask out reaches where it's thought that what is shown cannot represent the reality of the river flow, perhaps because it is 50% determined by a reservoir release or major diversion.

We agree that improving the representation of human controls on the hydrology is a key gap in global hydrological modelling and forecasting. We will highlight this more strongly in the revised manuscript and think it would best fit in "Sect. 4.4 Future directions". As per the response to your above comment, there is already layer called "Reservoir impact" within the GloFAS Web Map Viewer (Fig. B). We think this information is critical for forecast users as this layer informs the degree of control by reservoir operations on a hydrological forecast for a particular location, and therefore on the possible resulting increased uncertainty in the published forecasts.

[Figure]

**Fig. B.: Screenshot of the GloFAS Web Map Viewer (https://www.globalfloods.eu/) centered over Kenya with the "GloFAS Lakes and reservoirs" and "Reservoir Impact" layers activated, showing the example of the overall potential impact the Masinga reservoir has on the annual natural river discharge or the Tana River.**

Fig1: The label 'hydrological model' is interesting because most of the hydrological components of LISFLOOD are not used – really it is the empirical groundwater attenuation and the channel routing of runoff. A more descriptive label for this component might be 'Catchment and channel routing' (whether GW or channel it's all a kind of routing, conceptually). Not a big issue but it may be confusing given that a lot of hydrology (runoff generation, snow accum/melt, et) is done by the LSM. btw, LISFLOOD could conceivably also offer a hillslope routing function (gamma or UH distribution).

Thank you for this useful suggestion, this has been changed to 'Catchment and channel routing' in the resubmitted manuscript.

114: Would be helpful to add a sentence to clarify how these reservoirs are represented and their realism – eg level-pool scheme, fixed rule curve

We added the following sentence to the resubmitted manuscript: "Reservoir outflow is calculated with a set of four rules depending on the current reservoir filling level (see Burek et al., 2013)".

174: Perhaps add a sentence or two here or in the later discussion if the reanalysis latency impacts the skill of real-time forecasts (2-5 days is a lot of lag for a flood forecast!) or means that the skill of the hindcasts may be systematically higher than that of the real-time forecasts, since presumably the reanalysis initialization in the past doesn't have latency issues.

It is practically impossible to create identical forecast initialisation for reforecasts versus real-time forecast systems due to the constraint on availability of operational data streams – there will always be some lag. However, we do not simply leave this lag empty in GloFAS. We fill it up with the best estimate we have of real time conditions. In the case of the GloFAS 'fill up' (as shown in Figure 2 in the manuscript), the period from the last available GloFAS-ERA5 until real-time is based on the short-range (i.e. 1-day) ECMWF-ENS forecast control member and is only needed for the real-time forecast and not the reforecast.

178: Can you say what determines 'skillful'? eg 5% kge above benchmark in the SS?

Hydrological skill in Harrigan et al. (2020) is determined using the modified Kling-Gupta Efficiency Skill Score (KGESS) against a mean flow benchmark, with KGESS > 0 defined as skilful.

199, 235: Perhaps add a sentence explaining what fraction of these sites are impaired/ unimpaired and what 'synthetic' means in the context of an observation (eg labeled in the table). In general, it should be quite clear in the paper when you are verifying against the real versus perfect model world. The initial description of verification against in situ gage stations may slightly obscure the later default to benchmarking against the reanalysis discharge. Could a sentence be added at 235 to state what is gained/lost in the interpretation of skill through comparing to the perfect model benchmark?

In the Supplementary table, the label 'synthetic' under the 'Provider' column refers to a GloFAS diagnostic river point where no in situ observed time-series river discharge is available at this station location. These tables have now been updated to provide further metadata and are now publicly available as part of the "Documentation" together with the reforecast download on the CDS: https://cds.climate.copernicus.eu/cdsapp#!/dataset/cems-glofas-reforecast?tab=doc as well as added as Supplementary Table 1 in the resubmission.

Your comment regarding what is gained/lost calculating skill from the perfect model benchmark is similar to R#2. This approach is common practice in forecast evaluation, but for the benefit of a broader audience we have outlined the justification of this approach in Sect. 3 in the resubmitted manuscript (see response above to R#2), also mentioned below in response to your final comment.

239: again could you quantify in a sentence what you consider 'skillful'. without going fully statistical (ie significance level of a skill scores X% above 0), what is the rule of thumb used by the authors to consider a forecast 'skillful'?

As defined in L230-231, a CRPSS > 0 shows forecasts are more skilful than the benchmark.

Fig 5: It's curious that the persistence-benchmarked forecast SS is so high at timestep 1. If anything I would expect it to start slightly lower than its peak SS because as you move from longer leads to approach T0 the persistence forecast, in theory at least, approaches zero error, and the persistence and actual forecasts approach each other.

There must be something in practice here that means this is not the case, ie persistence has an offset at T0 that is not present in the actual forecast. Especially in medium to large rivers, much of the time by day 1 the flow from T0 has not changed much. Is it correct that the actual forecast is 95% better than persistence even at day1? Can/should this behavior be explained in the paper?

Your comments regarding interpretation of the forecast skill relative to the benchmark forecast is similar to that made by R#1. We found it useful to look at the CRPS values of individual components of the CRPSS skill score in Equation 1 to interpret the pattern as a function of lead time (Fig. A). While the CRPSS in Fig. 5 is dimensionless with an optimum value of 1, the CRPS error is measured in units of the variable being evaluated (here m$^3$/s) and so has an optimum of 0 (i.e. perfect accuracy against the reanalysis). It is the case that the persistence benchmark forecast is most accurate at a 1-day lead time and gets increasingly less accurate with longer lead times. But it is also the case that the GloFAS forecast is most accurate at a 1-day lead time with accuracy decaying with increasing lead time, but at a slower rate than the persistence benchmark, hence the CRPSS score comparing both CRPS.

To help users better interpret the CRPSS for their stations of interest, we have included CRPSS as well as CRPS plots as a clickable "pop out" window as part of the new GloFAS "forecast skill" layer on the GloFAS Web Map Viewer for the release of version 2.2 on 2 December 2020 (https://confluence.ecmwf.int/display/COPSRV/Latest+operational+release%3A+GloFAS+v2.2).

We have added Figure A as an additional panel - now Figure 5b in the resubmitted manuscript. To reflect the update to the CRPSS/CRPS plots within the "Forecast skill" layer on the GloFAS website, we have updated Sect. 4.2 and Fig. 9 in the resubmission.

[Figure]

**Figure A (New Figure 5b).: Global median Continuous Ranked Probability Score (CRPS) for GloFAS forecasts (black line) and both persistence (blue line) and climatology (red line) benchmark forecasts from 1- to 30-day lead times with respect to GloFAS-ERA5 river discharge reanalysis across 5997 diagnostic river points.**

343: I don't quite agree, as I think of bias & mean error-based scores reflecting only accuracy. The use of an integrated score such as crps means the results are sensitive not just to accuracy but to forecast spread (hence reliability). Perhaps rephrase here?

What we tried to say was that we focused on the overall correspondence between forecasts and the verifying reanalyses, and that there are many other aspects of forecast quality such as reliability (and spread) etc. that

could be unpacked in future evaluation work. This could be confused with the technical definition of the CRPS and so has been rephased to "The evaluation carried out here looks at the overall quality of forecasts".

354: It would be helpful to add a discussion returning to some of the major caveats on the applicability of the analysis. I strongly support the overall message of the paper and commend the effort for having generated reanalyses and used them to demonstrate the kind of information one can derive from them, but I also recognize that in many areas the skill / usability estimates are compromised by use of perfect model benchmarks, and the lack of representation of real-world impairments in GLOFAS. Could the authors walk us through some of the possible limitations, eg in a paragraph, discuss a few cases that users may face in interpreting this kind of skill or using the reforecasts? eg, in the best case, through the model may have some biases it generally represents catchment variability such that the perfect model skill analysis is more or less directly transferrable to real world conditions. A use could infer usability qualitatively or even apply quantitative post processing methods for their own site observations to adjust the skill scores accordingly. In a medium case, the model is badly biased (say magnitudes by 50-100% and seasonal timing off by 30 days), but still represents observed variability in high/low flow conditions – what more would be required of a user then? And in the worst case, say on highly regulated rivers, perhaps glofas cannot be used except in the most extreme situations, eg when there is so much or so little water that management effects are secondary. Would this suggest any other future directions, eg providing guidance or tools on such user-based post-processing and analysis?

Reviewer 2 had a similar comment on highlighting the justification and caveats of using proxy observations in the evaluation. We have added the justification of this approach at the end of Sect. 3.3 in the resubmitted manuscript as follows:

"Calculating forecast skill against proxy observations such as reanalysis is common in hydrological forecasting as it has the advantage of providing a spatiotemporally complete picture of forecast skill, currently not possible based on availability of the current global in situ observed river network (Lavers et al., 2019). It also allows the forecast predictability range to be isolated in the absence of systematic hydrological model errors. There is a disadvantage of forecast evaluation against proxy observations for catchments that represent hydrological dynamics poorly. While Harrigan et al. (2020a) demonstrate the performance of GloFAS-ERA5 reanalysis is largely hydrologically skilful, readers should be aware that there are areas where performance is poor and that there are large parts of the world where the performance is unknown due to the lack of in situ observations to evaluate against (Figure 4)."

As GloFAS does not apply any post-processing, there is indeed a lot of room for users to increase the forecast quality if they carry out a further post-processing step at their end – this is a very good point and highly encouraged, we have therefore added the following to Sect. 4.4:

"GloFAS forecasts and reforecasts have not been post-processed, therefore there is room for users to increase further forecast quality by applying post-processing with their local observations data to correct forecast bias or timing errors, for example."

Your last point is probably one of the biggest challenges in large-scale hydrological forecasting. While we can provide information on whether GloFAS is on average skilful or not for a particular location, this becomes more challenging in areas with very little information on river management and/or a lack of situ observations. However, when faced with an emergency situation due to very extreme hydrometeorological conditions, for example tropical cyclone Idai that devastated Mozambique in March 2019, operational experience has shown that despite the uncertainty associated with GloFAS forecasts, useful information on the future evolution of flood risk can be made, when balanced with many additional sources of information (Emerton et al., 2020).

Thank you again for your insight and we appreciate your time to review our manuscript,

Kind regards,
Shaun Harrigan on behalf of all co-authors

**References**

Alfieri, L., Burek, P., Dutra, E., Krzeminski, B., Muraro, D., Thielen, J., and Pappenberger, F.: GloFAS – global ensemble streamflow forecasting and flood early warning, Hydrol. Earth Syst. Sci., 17, 1161–1175, https://doi.org/10.5194/hess-17-1161-2013, 2013.

Alfieri, L., Pappenberger, F., Wetterhall, F., Haiden, T., Richardson, D. and Salamon, P.: Evaluation of ensemble streamflow predictions in Europe, J. Hydrol., 517, 913–922, doi:10.1016/j.jhydrol.2014.06.035, 2014.

Burek, P., van der Knijff, J. M. and de Roo, A. P. J. D.: LISFLOOD - Distributed Water Balance and Flood Simulation Model - Revised User Manual, Publications Office of the European Union, doi: 10.2788/24719, 2013.

Emerton, R., Zsoter, E., Arnal, L., Cloke, H. L., Muraro, D., Prudhomme, C., Stephens, E. M., Salamon, P. and Pappenberger, F.: Developing a global operational seasonal hydro-meteorological forecasting system: GloFAS-Seasonal v1.0, Geoscientific Model Development, 11(8), 3327–3346, https://doi.org/10.5194/gmd-11-3327-2018, 2018.

Emerton, R., Cloke, H., Ficchi, A., Hawker, L., de Wit, S., Speight, L., Prudhomme, C., Rundell, P., West, R., Neal, J., Cuna, J., Harrigan, S., Titley, H., Magnusson, L., Pappenberger, F., Klingaman, N. and Stephens, E.: Emergency flood bulletins for cyclones Idai and Kenneth: A critical evaluation of the use of global flood forecasts for international humanitarian preparedness and response, Int. J. Disaster Risk Reduct., 101811, doi:10.1016/j.ijdrr.2020.101811, 2020.

Lavers, D., Harrigan, S., Andersson, E., Richardson, D. S., Prudhomme, C., and Pappenberger, F.: A vision for improving global flood forecasting, Environ. Res. Lett., https://doi.org/10.1088/1748-9326/ab52b2, 2019.

Harrigan, S., Zsoter, E., Alfieri, L., Prudhomme, C., Salamon, P., Wetterhall, F., Barnard, C., Cloke, H. and Pappenberger, F.: GloFAS-ERA5 operational global river discharge reanalysis 1979–present, Earth Syst. Sci. Data, 12(3), 2043–2060, doi:https://doi.org/10.5194/essd-12-2043-2020, 2020.

Haiden, T., Janousek, M., Vitart, F., Ferranti, L., Prates, F. and Prates, F.: Evaluation of ECMWF forecasts, including the 2019 upgrade, 2019.

Knoben, W. J. M., Freer, J. E., and Woods, R. A.: Technical note: Inherent benchmark or not? Comparing Nash–Sutcliffe and Kling–Gupta efficiency scores, Hydrol. Earth Syst. Sci., 23, 4323–4331, https://doi.org/10.5194/hess-23-4323-2019, 2019.

Pechlivanidis, I. G., Crochemore, L., Rosberg, J., and Bosshard, T.: What are the key drivers controlling the quality of seasonal streamflow forecasts? Water Resources Research, 56, e2019WR026987. https://doi.org/10.1029/2019WR026987, 2020.

Zajac, Z., Revilla-Romero, B., Salamon, P., Burek, P., Hirpa, F. A. and Beck, H.: The impact of lake and reservoir parameterization on global streamflow simulation, J. Hydrol., 548, 552–568, doi:10.1016/j.jhydrol.2017.03.022, 2017.

---

## Author Response (AR2)

Dear Jim,

Please find below our responses to the three reviewers. We highlight where we have made changes to the resubmitted manuscript considering their comments.

Kind regards,
Shaun Harrigan on behalf of all co-authors

**Response to R#1**

*Anonymous Referee #1 (R#1)'s original text in* black *with our response in* blue*.*

My apologies to the editor and the authors for the lateness of this review. Thanks very much to the authors for comprehensively addressing the first of my objections relating to CRPSS calculated with respect to persistence. It's clear from their explanation and additional figure that this was not an error, and I appreciate the detailed explanation and figure. (We can disagree on whether this result is 'intuitive' or not - in the responses the authors stated that the decline in accuracy of persistence is faster than GloFAS, which is what I (intuitively) assumed would happen, but it's clear this is not the case as skill falls with lead time. But this is basically irrelevant.) On re-reading the description of persistence skill (L234-235) there was one thing that was a little ambiguous to me - if a forecast is issued on Jan 2, where the first forecast value is for Jan 3, is the persistence forecast taken from Jan 1 or Jan 2? (I had assumed Jan 2, but I wasn't sure from their description.) This may be worth clarifying.

We are glad our previous explanation and figure have cleared up your previous concern on the CRPSS.

To avoid any ambiguity in the description of the persistence benchmark forecast we have added how the persistence forecast is calculated for an example forecast issued on 3 January. The following text was added to L235-238 in the revised manuscript:

"For example, for a forecast issued on 3 January at 00UTC, the persistence benchmark forecast is the average river discharge over the 24 h time step from 2 January 00UTC to 3 January 00UTC, and the same value is used as benchmark for all 30 lead times (i.e., 4 January to 2 February)."

Added to this is that the many strengths of the paper remain: GloFAS is a system of international significance and this update is in my view ultimately absolutely worthy of publication. The paper is very clear and exceptionally well presented, and remains a join to read.

We thank the reviewer for their positive words and recommendation.

However, the authors chose not to address my second major comment about reliability (and I note they have the support of the editor in this, so I am probably howling into the wind here). Essentially I disagree with both the authors and the editor that omitting an assessment of reliability is acceptable. To me there are two fundamental reasons to present ensemble predictions (and I'm not alone here, going back at least as far as Krzystofowicz 2001): 1) The predictions are, on average, more accurate than deterministic predictions and 2) the forecasts are more 'honest' because they give a representation of predictive uncertainty.

The authors have clearly and admirably addressed (1) with their analysis of CRPS. The second they have ignored. I note also that Emerton et al. (2018) presented a basic assessment of reliability in their assessment of GloFAS-seasonal - I don't understand why such an analysis could not be performed here. (Though as I stated previously - a summary statistic of PIT values would be more appropriate, though attributes diagrams would be ok.) For the addition of one figure and perhaps a paragraph or two, the authors could (and in my view

should) have addressed this fundamental aspect of ensemble prediction. If the authors were concerned about having too many figures, in my view they could remove Fig 11, which is unlikely to be read in detail and (presumably) can be accessed through the GloFAS portal.

The aim of our paper is not a full-scale evaluation of GloFAS forecast skill for a comprehensive range of forecast aspects. Instead, the primary aim is to present to the community the model components and configuration used to generate operational global-scale forecasts and reforecasts. A secondary aim is to establish the science underpinning a new 'headline skill score' that is made available to users on the GloFAS Web Map Viewer (i.e. Figure 9). Our choice of CRPS as the metric was based on Pappenberger et al. (2015), among many others, who state that the most well-known overall summary metric used for a 'headline score' in operational ensemble forecasting is the CRPS. We also note that the CRPS, as an overall metric, does consider reliability implicitly; the CRPS is penalised for ensemble forecasts with lower reliability compared with forecasts with higher reliability (Hersbach, 2000). To comprehensively attribute which forecast aspects (there are many other than reliability) are leading to higher or lower overall forecast skill at locations under different hydroclimate regimes, times of year, etc., is a stand-alone paper itself.

Therefore, we strongly believe such an analysis does not fit into the current scope of our paper and standby our original comment. Future work to expand the number of evaluation metrics offered through GloFAS is underway, and we fully agree with and thank you for your suggestion to ensure reliability information is part of this work going forward, but we defend our current choice of CPRS (and hence CRPSS) as the GloFAS headline skill score in this first step.

**References**

Hersbach, H.: Decomposition of the Continuous Ranked Probability Score for Ensemble Prediction Systems, Wea. Forecasting, 15, 559–570, https://doi.org/10.1175/1520-0434(2000)015<0559:DOTCRP>2.0.CO;2, 2000.

Pappenberger, F., Ramos, M. H., Cloke, H. L., Wetterhall, F., Alfieri, L., Bogner, K., Mueller, A., and Salamon, P.: How do I know if my forecasts are better? Using benchmarks in hydrological ensemble prediction, Journal of Hydrology, 522, 697–713, https://doi.org/10.1016/j.jhydrol.2015.01.024, 2015.

**Response to R#4**

*Anonymous Referee #4 (R#4)'s original text in* black *with our response in* blue*.*

This is a review of "Daily ensemble river discharge reforecasts and real-time forecasts from the operational Global Flood Awareness System". I was asked to review this paper following a first round of revisions and have had the opportunity to evaluate the author responses to previous reviewers' comments. I think the authors have provided a thorough and convincing revision to these comments in the submitted manuscript.

As for my own personal review: I have read the paper and I think that this paper should be published promptly. This advancement seems to me as a fantastic contribution to the hydrological sciences community. I applaud the endeavor and think that HESS is a suitable journal for this type of contribution. I think the science (GloFAS model evaluation methods, figures and description of available forecast datasets) is sound and the end-product will be useful to many researchers. The structure is clear, concise and the text very well written.

We thank the reviewer for taking the time to review our paper, and especially going through previous review comments and our revisions, and for your positive words. Much appreciated.

I have but a few comments that I think the authors can easily address and which can be handled at the editorial board level.

1- Line 41: Australian --> Australia

Now changed in the revised manuscript

2- Lines 56-57: Here the word "global/globe" is used 3 times in the span of 21 words. I suggest varying a bit as to not distract the reader.

Now modified to: "GloFAS can be used for providing daily assessments of potential upcoming flood events for the whole globe, such a spatio-temporal consistent overview is required by several users" in the revised manuscript.

3- Line 91: Reference to a future date that is actually in the past. Actually, most reforecast dates, IFS model cycles, versions, etc. need to be updated. I acknowledge that this is due to the review process, however I just want to highlight that this should be updated prior to publication.

We have now been through the manuscript and have updated all necessary dates, links, and references to model versions such as the IFS that have advances since the paper was first submitted. Note that the reforecast dates are the same.

4- General question: GloFAS-ERA5 uses ERA5 since it is updated near-real-time. However, ERA5-Land will soon also be near-real-time and will have the same native resolution as GloFAS (although the authors are better placed than I am to talk about this!). I wonder if there is any plan to update GloFAS-ERA5 to GloFAS-ERA5Land once it becomes near-real-time? If so, perhaps add a sentence on this in the concluding remarks or discussion?

Yes indeed, we have already assessed the potential gains in moving to ERA5-Land and if you are interested have an initial experiment for river discharge published within Muñoz-Sabater et al. (2021) whereby we benchmark GloFAS-ERA5 against GloFAS-ERA5-Land at 1285 river discharge stations – spoiler is that ERA5-Land does show moderate improvements over ERA5 in the majority of catchments.

**Reference**

Muñoz-Sabater, J., Dutra, E., Agustí-Panareda, A., Albergel, C., Arduini, G., Balsamo, G., Boussetta, S., Choulga, M., Harrigan, S., Hersbach, H., Martens, B., Miralles, D. G., Piles, M., Rodríguez-Fernández, N. J., Zsoter, E., Buontempo, C., and Thépaut, J.-N.: ERA5-Land: A state-of-the-art global reanalysis dataset for land applications, 1–50, https://doi.org/10.5194/essd-2021-82, 2021.

Congratulations once again on the, to me, excellent paper.

Thank you very much again.

**Response to R#5**

*Anonymous Referee #5 (R#5)'s original text in* black *with our initial response in* blue.

--- Summary

The manuscript by Harrigan et al. was in its second round of reviews when I first reviewed it. The paper is of high quality and will be undeniably valuable to future users of the openly available GloFAS hindcasts. My

review takes into account the replies of the authors to the previous reviews, replies which I found thorough and well supported. Nevertheless, after reading the manuscript, I still had questions similar to some of the comments previously raised, and therefore think some of these well justified choices deserve to be mentioned, even briefly, in the manuscript. Hereafter, I list some recommendations for improving explanations, some minor points, as well as a concern about Figure 5.

We thank the reviewer for taking the time to review our paper, and especially going through previous review comments and our revisions, and for your positive words. Much appreciated.

--- General comments

The authors have replied to comments made by previous reviewers on the subject, but I believe there should be in the paper an explanation for why the evaluation of a flood awareness system is not based on its capacity to forecast floods.

We stand by our decision to choose the CRPS (and CRPSS) as choice of headline skill score, it is the most well-known and scientifically validated in the operational ensemble forecasting literature, as we highlight in L248-250. It was critical that we have such a headline skill score that can be used to track overall baseline ensemble skill against persistence and climatology. This is essential information for forecasters to understand the fundamental scientific quality of GloFAS in their area of interest. It must also be noted that GloFAS is not only just used for flood forecasting, the ensemble forecasts cover the full flow regime and GloFAS has seasonal products for both high and low flow (https://confluence.ecmwf.int/display/COPSRV/GloFAS+Hydrological+Products+Overview). As mentioned in Sect. 4.4, this is just assessment of overall forecast skill and we highlight (L417-423) that future work should indeed look at other aspects of forecast quality, such as performance during extremes.

I wish the authors would better explain their choice of changing benchmark depending on the lead time, the reasoning behind the choice of the 0.5 threshold, and the expected influence on the results presented in Sections 4.2.2 and 4.2.3. For instance, why not choose the toughest benchmark to beat, designing for each catchment a benchmark based on both persistence and climatology that would resemble a locally optimized system switching from persistence to climatology when the CRPS of persistence and climatology cross?

There is some guidance in the literature on which is the most appropriate benchmark to use, depending on the lead time. We cite Pappenberger et al. (2015) in L232-236 who advise persistence benchmarks for short and medium range lead times and climatology benchmarks for longer lead times. GloFAS covers all three ranges from short- (1-3 days), medium (4-15 days) and extended (from 15-30 days), therefore there is no single most appropriate benchmark for all lead times assessed. There was a lack of guidance in the literature on out to exactly which lead time is persistence the most appropriate benchmark, and when climatology should be used from for global scale forecast models. This is exactly the analysis presented in Figure 5. We produce the experiment against both persistence and climatology for all lead times, and so can find the lead time, when averaged for all stations across the globe, at which persistence and climatology should be used. This result then informs the choice of benchmark in Section 4.2.2 and 4.2.3, which we believe is well justified from Figure 5.

We thought it best in Figure 6 and Figure 7 to not blend benchmarks when assessing the spatial distribution of skill. Essentially, adding a blended benchmark comprised of both persistence and climatology would add an additional degree of freedom in terms of interpreting the results. For example, is the GloFAS forecast skill different between two catchments because the one uses climatology and one uses persistence?

However, for the purpose of the new headline skill score layer on the Web Map Viewer (i.e. Section 4.2 & Figure 9), we do consider both persistence or climatology independent of lead time. If for example the CRPSS

against a climatology forecast drops below the 0.5 'high skill' headline score threshold at a 2 day lead time, and skill against persistence remains above, then we define the station is only 'highly skilful' out to 2-days.

Regarding the choice of CRPSS = 0.5 for our threshold, we categorise forecasts above this threshold as 'highly skilful' compared to the benchmark (i.e. GloFAS is 50% more accurate than the benchmark). There is no 'correct' choice of threshold, it is arbitrary, but if too low then on the map (Fig. 9) all stations would appear 'highly skilful' and would not be useful for users to see quickly at a global view, which regions tend to have higher or lower skill. Once a user clicks on an individual station then we show the raw CRPSS and CRPS for against both benchmarks, thus giving users all the information to make their own assessments.

I also have some concerns about the maps and some statements presented in Section 4.2.2. More specifically, to which extent does your choice of benchmark influence these spatial patterns? You chose to select the benchmark based on the lead time. Therefore, when looking at spatial skill patterns for lead times shorter than 15 days, won't the highest skills tend to be found in catchments where persistence is not adequate, i.e. catchments where seasonality dominates (hence the low skill in polar latitudes), rather than in places where GloFAS hindcasts are indeed of good quality? Reversely, at lead times longer than 15 days, the highest skills should be found in catchments where serial correlation dominates.

See our comments above, you are exactly right: but we think blending the benchmarks would make it even more difficult to understand the differences in skill (is it due to poorer forecast skill, strong serial correlation OR a climatologically stable location).

You mention L.294-295 that "The regions of highest skill are similar to those for short- and medium-range", but this is not obvious when comparing areas without skill in Figure 6d and Figure 7a and it seems to me that what we observe are artifacts from this strict split in benchmark at 15 days lead. For instance, parts of Russia go from negative skills at lead time 10 to positive skills at lead time 15 days. This is counter-intuitive, and I think a comment from the authors would be necessary.

This is an interesting point and yes is due to the choice of benchmark. We fully agree this deserved a comment in the manuscript, thank you for highlighting this fascinating example. We have therefore added the following paragraph to L305-311 in the revised manuscript:

"The choice of benchmark forecast used for short- to medium-range (i.e., Figure 6) and extended-range (i.e., Figure 7) maps was based on the global median of all stations in Figure 5a. However, there is spatial variability in the choice of best benchmark according to unique hydroclimate properties. For example, in northern latitudes around Russia and northern Scandinavia GloFAS was shown to be negatively skilful against persistence at a 10-day lead time (Figure 6d). However, GloFAS is shown to be skilful against a climatology benchmark in the same region at lead time 15 days. This shows that persistence is a much tougher benchmark to beat in these catchments compared to climatology, likely due to the high degree of serial correlation from snow processes"

Finally and in line with these comments, I question the use of "should be" L.432-433 in the conclusion: "The analysis shows that on average across the world forecasts should be benchmarked against persistence for lead times up to about two weeks and against climatology for longer lead times." It currently sounds as a recommendation, but it seems that this analysis should be performed and fine-tuned for use at non-global scale.

We agree that this sentence sounds like a recommendation and is not the main intention of the analysis, we have therefore removed this sentence.

In several of your replies to R2, you mentioned that a novelty from this work was that the reforecasts, along with other data types, are open. Opportunities for further research based on this data are well mentioned, but I expected a deeper discussion on the incentives for opening up and on encountered challenges. I believe your experience may be valuable for other data providers choosing to switch to open access, and highlighting overcome challenges would only highlight the value of this work.

We had not considered this point but are very happy to share our experience if it could be useful for other groups. We have therefore added the following paragraph to Section 4.3 L411-427 in the revised manuscript:

"While producing large sets of reforecasts and providing data free and open to the community has many benefits, it comes with challenges and key considerations. One of the main considerations is the data storage and delivery infrastructure. A full set of 20-year GloFAS reforecasts is ~23 TB in size. For each new major model upgrade a new set of reforecasts are generated, together with ~35 GB of raw data generated every day for the real-time forecast stream. It is clear that the size of data is a barrier for many users to use. Most users do not require the full temporal range of data and are usually interested in a sub-domain, for example their study region or country. It is simply not practical for every user to download ~23 TB of global data to their computing infrastructure if they only want data for their individual catchment, not to mention if a standard laptop is the only computer available to them. Our solution was to store GloFAS data on the ECMWF Meteorological Archival and Retrieval System (MARS; https://confluence.ecmwf.int/display/UDOC/MARS+user+documentation, last accessed: 25 March 2022) – MARS offers the functionality for users to choose temporal and/or spatial subsets (among others) and the heavy data handling and computation happens on ECMWF infrastructure so the user can download a smaller and more manageable subset of data. The CDS is the public facing front end for users to access GloFAS data and metadata, and communicates with MARS in the backend. A further consideration is producing sufficient documentation for users to interact with the data and provision of a support service whereby users can get in contact with GloFAS data and domain experts for queries: https://confluence.ecmwf.int/site/support (last accessed: 25 March 2022)."

There seems to be an inconsistency between Figures 5a and 5b. It is unclear why the intersection between the median CRPS of persistence and the median CRPS of climatology (~18 days) does not correspond to the intersection of the median CRPSS (~14 days). I could not find any reasonable reason why this would happen.

They are not expected to be the same. In Fig. 5b the median CRPS of persistence (blue line) and median CRPS of climatology (red) line cross at day 18, but this does not factor in the CRPS of the GloFAS forecast (black line), whereas in Fig. 5a, the **CRPSS** is expressed using the CRPS of GloFAS AND both persistence and climatology benchmarks.

--- Minor comments

L.14 Can we really say that global-scale hydrological forecast systems are widely used ?

Have removed "widely" in the revised manuscript.

L. 80-81 'reforecasts' (as ?) consistent as possible

Have added 'as' in the revised manuscript.

L.92 'of with' seems strange

Changed to "with increased data access availability" in the revised manuscript.

L.140-143 The parenthesis count is off.

Now fixed.

L.165 The second incentive for using hindcasts is not as universal as incentives 1) and 3). In fact, the reader only links IFS updates and influence on the hindcasts in the next paragraph. I suggest moving incentive 2) later in the text, and use this information not as a common incentive to using hindcasts, but rather as additional information on the choice of 2019 as reference year for the system versions.

Done.

Figure 3 For the sake of showing a self-explanatory figure, I suggest explaining « ens=11 » in the caption or explicitly writing « 11 members ».

Updated the caption as suggested.

Figure 4 KGESS should be briefly explained when presenting this figure. The authors offered a short and clear explanation in their reply to previous comments which could very well fit in the paper and ease understanding.

We realised we did not add the abbreviation of KGESS, now added. You can see where the KGESS is explained from L199-204 in the revised manuscript.

L.271-272 "At day 15…" This sentence is unclear.

Now reads "At day 15 the CRPSS is…" in the revised manuscript.

L.276 "The aid the readers…" seems incorrect. Did you mean "To aid the readers…"?

Yes, thank you. Now corrected.

L.287 "across at" sounds strange.

"at" has been removed.

Figure 8a Please add an explanation in the caption for the red and blue-shaded areas.

Now reads: "…at 5997 diagnostic river points by degree latitude of the river point with polar (tropics) climate region shaded in blue (red) (a)"

L. 352 Remove "be"

Done.

L.434 "… results show…"

Done.